# Connecting MHC-I-binding motifs with HLA alleles via deep learning

Ko-Han Lee [1], Yu-Chuan Chang[1], Ting-Fu Chen[1], Hsueh-Fen Juan [1,2,3,4], Huai-Kuang Tsai [1,5] & Chien-Yu Chen [1,6✉]

The selection of peptides presented by MHC molecules is crucial for antigen discovery. Previously, several predictors have shown impressive performance on binding affinity. However, the decisive MHC residues and their relation to the selection of binding peptides are still unrevealed. Here, we connected HLA alleles with binding motifs via our deep learning-based framework, MHCfovea. MHCfovea expanded the knowledge of MHC-I-binding motifs from 150 to 13,008 alleles. After clustering N-terminal and C-terminal sub-motifs on both observed and unobserved alleles, MHCfovea calculated the hyper-motifs and the corresponding allele signatures on the important positions to disclose the relation between binding motifs and MHC-I sequences. MHCfovea delivered 32 pairs of hyper-motifs and allele signatures (HLA-A: 13, HLA-B: 12, and HLA-C: 7). The paired hyper-motifs and allele signatures disclosed the critical polymorphic residues that determine the binding preference, which are believed to be valuable for antigen discovery and vaccine design when allele specificity is concerned.

[1] Taiwan AI Labs, Taipei 10351, Taiwan. [2] Graduate Institute of Biomedical Electronics and Bioinformatics, National Taiwan University, Taipei 10617, Taiwan. [3] Department of Life Science, National Taiwan University, Taipei 10617, Taiwan. [4] Center for Computational and Systems Biology, National Taiwan University, Taipei 10617, Taiwan. [5] Institute of Information Science, Academia Sinica, Taipei 11529, Taiwan. [6] Department of Biomechatronics Engineering, National Taiwan University, Taipei 10617, Taiwan. ✉email: chienyuchen@ntu.edu.tw

Antigens are essential for the induction of adaptive immunity to respond to threats, such as infectious diseases or cancer[1]. Most antigens are short non-self-peptides; however, not all peptides are antigenic[1]. Researchers have been committed to the development of peptide-based vaccines to prevent or treat numerous diseases[2–5]. For instance, tumor neoantigens, derived from proteins with nonsynonymous somatic mutations, may be suitable cancer therapeutic vaccines[6–8]. In order to choose good antigens, it is important to understand the process of antigen presentation.

Major histocompatibility complex class I (MHC-I) molecules are cell surface proteins essential for antigen presentation[1]. MHC-I encoded by three gene loci (HLA-A, -B, and -C) are composed of a polymorphic heavy α-chain and an invariant β-2 microglobulin light chain[9]. The α1- and α2-domains form the peptide-binding cleft, a highly polymorphic region, contributing to the diversity of MHC-I-binding motifs[9]. There are >13,000 MHC-I alleles on a four-digit level (e.g., A*02:01) recorded in the IPD-IMGT/HLA database[10], representing a particular protein sequence. Thus, it is difficult to select antigens from numerous peptides for each MHC allele via experiments.

In order to facilitate the process of antigen discovery, several predictors have been developed and shown accurate performance on MHC-I–peptide binding affinity[11,12]. Owing to the similarity of polymorphic regions in MHC-I alleles, researchers tended to build a single pan-allele predictor rather than numerous allele-specific predictors[13]; of note, a pan-allele predictor takes both MHC-I and peptide sequences as the input. A pan-allele predictor is thought to disclose the connection among different alleles via the consensus pattern in polymorphic regions[13]. Nevertheless, the relation between MHC-I sequences and their binding motifs is still unspecified.

In the past years, a few studies have discussed the similarity between MHC-I-binding motifs[14–16]. Some key residues of MHC-I molecules determine the binding motifs that can be clustered into several groups[14]; the types of key residues within allele clusters and motif clusters are consistent to some extent[15]. In addition, the similarity between binding motifs can be used to improve the performance of binding prediction[16]. However, it is difficult to specify the key residues of each motif group from the limited number of alleles with experimental measurements.

In this regard, we developed a deep learning-based framework, MHCfovea, that incorporates supervised binding prediction with unsupervised summarization to connect important residues to binding preference. As exemplified in Fig. 1, this study explored the binding potential of billions of peptide–allele pairs via the prediction module; only qualified binding pairs were sent to the summarization module to infer the relation between binding motifs and MHC-I sequences. In the end, the resultant pairs of hyper-motifs and allele signatures can be easily queried through a web interface (https://mhcfovea.ailabs.tw).

## Results

### Overview of MHCfovea.
MHCfovea integrates a supervised prediction module and an unsupervised summarization module to connect important residues to binding motifs (Fig. 1). The predictor in the prediction module is constructed of an ensemble model based on convolutional neural networks (CNN) (Supplementary Fig. 1) embedded with ScoreCAM[17], a class activation mapping (CAM)-based[18] approach, to highlight the important positions of the input MHC-I sequences. As for the summarization module, to infer the relation between the important residues and the binding motifs, we made predictions on unobserved alleles to expand our knowledge from 150 to 13,008 alleles followed by clustering all N- and C-terminal binding motifs, respectively. Then the corresponding signatures of MHC-I sequences on the important positions were generated to reveal the relation between MHC-I sequences and their binding motifs. In the following subsections, we first demonstrate the performance of MHCfovea's predictor using 150 alleles with experimental data. Second, we introduce the important positions highlighted by ScoreCAM embedded in MHCfovea's predictor. Finally, we present the summarization results on 13,008 alleles in the groups of HLA-A, -B, and -C, respectively. Additionally, alleles from the same HLA group but falling into different clusters are identified to disclose the critical residues that determine the binding preference beyond the HLA groups.

### Performance evaluation of MHCfovea's predictor.
The predictor of MHCfovea takes an MHC-I-binding cleft sequence with 182 amino acids (a.a.) and a peptide sequence with 8–15 a.a.[19] to predict the binding probability. We trained the predictor using 150 alleles with either binding assay data or ligand elution data and then tested it on an independent ligand elution dataset built by Sarkizova et al.[15]. We adopted a large number of in silico decoy peptides in parallel with in vivo free peptides (not present on MHC-I molecules) to train and test the predictor; of note, we took NetMHCpan4.1[20] as a reference to set the ratio of decoy peptides to eluted peptides (decoy-eluted ratio (D-E ratio)) at 30 in the benchmark (testing) dataset. The data sources used are characterized in Supplementary Table 1 and Supplementary Data 1.

The number of decoy peptides is notably higher than that of eluted peptides, meaning that MHC-I–peptide binding prediction is an extremely imbalanced classification process. In fact, the imbalance among classes is a common issue in machine learning, and some methods have been developed to deal with it[21]. In MHCfovea, we used the ensemble strategy with downsampling[22–24] to resolve such an imbalanced learning task (Fig. 2a).

Next, to evaluate the effect of the D-E ratio in the overall training dataset (denoted as $A$ in Fig. 2a) and the D-E ratio in each downsized dataset (denoted as $B$ in Fig. 2a), we trained models with five different D-E ratios ($B = 1, 5, 10, 15,$ and 30) in each downsized dataset and three different D-E ratios ($A = 30, 60,$ and 90) in the training dataset. Of note, all experimental data were shared in each downsized dataset, and the decoys were non-overlapping between each downsized dataset to make sure all the decoys were used in the ensemble model eventually. Figure 2b depicts the performance of the validation dataset (Supplementary Tables 2 and 3). The best model was with D-E ratios of $B = 5$ and $A = 90$, showing an average precision (AP) of 0.898 and an area under the receiver operating characteristic (ROC) curve (AUC) of 0.991. Therefore, we used the ensemble model with 18 (=90/5) CNN models (the best performance on the validation dataset) as the predictor of MHCfovea.

To compare MHCfovea's predictor with other well-known predictors, including NetMHCpan4.1[20], MHCflurry2.0[25], and MixMHCpred2.1[16], we adopted an independent benchmark dataset from NetMHCpan4.1. Even though the testing data (benchmark) are the same in the comparison of this study, the training data of different predictors are not consistent. Supplementary Table 4 and Supplementary Fig. 2 summarized the training dataset used by each predictor. Both of NetMHCpan4.1 or MHCflurry2.0 used more alleles and a larger number of experimental measurements (positive peptides) than MHCfovea. To be specific, only one peptide is unique in MHCfovea (Supplementary Fig. 2b). As for MixMHCpred2.1, MHCfovea as well as the other two predictors used more alleles and peptides than it owing to the paucity of public data when MixMHCpred2.1 was published.

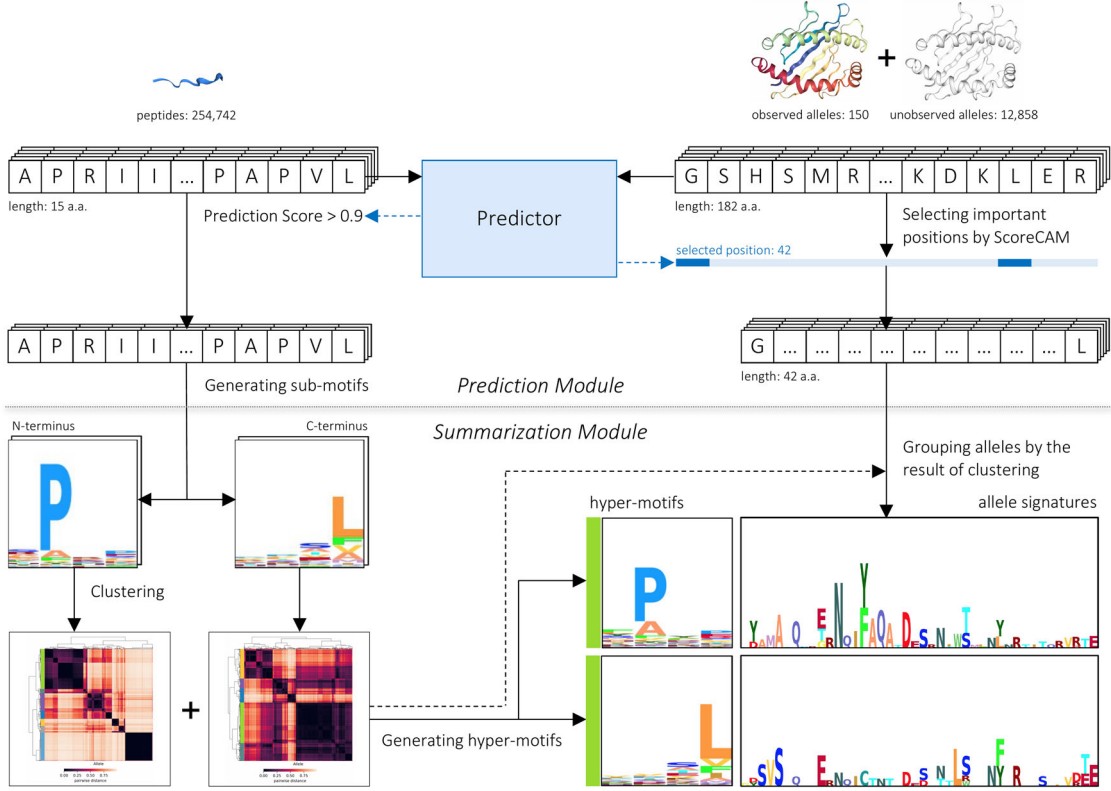

**Fig. 1 An overview of MHCfovea.** MHCfovea, a deep learning-based framework, contains a prediction module and a summarization module that infers the relation between MHC-I sequences and peptide-binding motifs. First, the predictor, an ensemble model of multiple convolutional neural networks (CNN models), was trained on 150 observed alleles. In the predictor, 42 important positions were highlighted from MHC-I sequence (182 a.a.) using ScoreCAM. Next, we made predictions on 150 observed alleles and 12,858 unobserved alleles against a peptide dataset (number: 254,742) and extracted positive predictions (score >0.9) to generate the binding motif of an allele. Then, after clustering the N-terminal and C-terminal sub-motifs, we built hyper-motifs and the corresponding allele signatures based on 42 important positions to reveal the relation between binding motifs and MHC-I sequences.

On the benchmark dataset, MHCfovea showed an AUC of 0.977 (Fig. 2c and Supplementary Table 5) and an AP of 0.841 (Supplementary Fig. 3a and Supplementary Table 5), both better than those obtained with the other predictors. The primitive output of MHCfovea is the estimated probability of allele–peptide binding. For the threshold of predicting an input pair as positive, setting a threshold at 0.68 reaches a maximal F1 score of 0.837 on the validation dataset. This threshold is suggested when adopting MHCfovea as a binary predictor. Apart from the whole benchmark dataset, we also evaluated the performance on every allele. MHCfovea showed a median AUC value of 0.984. For 82 of the 92 (89%) alleles, the AUC is at least 0.95. MHCfovea performed significantly better than the other predictors with respect to the AUC and AP metrics (Fig. 2d, Supplementary Fig. 3c, and Supplementary Data 2).

Next, the performance of our pan-allele model was carefully examined in the context of 16 unobserved alleles (with no experimental measurements in the training dataset), listed in Supplementary Table 6. Importantly, there is no significant difference between the AUC and AP of unobserved alleles and of the observed alleles (Fig. 2e, Supplementary Fig. 3e, and Supplementary Data 3), suggesting that MHCfovea shows good performance not only toward alleles present in the training data but also in the context of unobserved alleles. Furthermore, when compared with other predictors on the ten commonly unobserved alleles across all the predictors, listed in Supplementary Table 6, MHCfovea also has slightly better performance (Supplementary Fig. 3f, g and Supplementary Data 3). The high similarity of sequences between alleles in the same HLA group was regarded as

a reason for the good performance on unobserved alleles. Nevertheless, B*55:02 is an unobserved allele with an AUC of 0.993, while no alleles in the group B*55 are present in the training dataset, giving an example of MHCfovea's good accuracy on the alleles of a rarely observed HLA group.

To further evaluate the reliability of the MHCfovea's predictor on unseen peptides, we took the sets of similar and dissimilar peptides in the benchmark dataset into consideration, where similar peptides denote a peptide in the testing data is identical or near-identical (one peptide is another peptide's substring) to any peptides in the training or validation data. Because most experimental data were conducted on normal human cells, it is possible to have identical or near-identical peptides in the benchmark dataset even when we require that no identical allele–peptide pairs are present in the benchmark and training (or validation) data simultaneously. Finally, benchmark data were partitioned into four groups (Supplementary Table 7): (1) unobserved alleles paired with dissimilar peptides; (2) unobserved alleles paired with similar peptides; (3) observed alleles paired with dissimilar peptides; and (4) observed alleles paired with similar peptides. Figure 2f and Supplementary Fig. 3h (Supplementary Data 4) provide the results on the metrics of AUC and AP, respectively. For each group, MHCfovea outperformed the other predictors in the respect of AUC and has better AP than MHCflurry2.0 and MixMHCpred2.1. Undeniably, similar peptides have better performance than dissimilar peptides in MHCfovea, and this phenomenon did not appear in other predictors because the definition of similar and dissimilar peptides might not applicable on them because the training data of each predictor

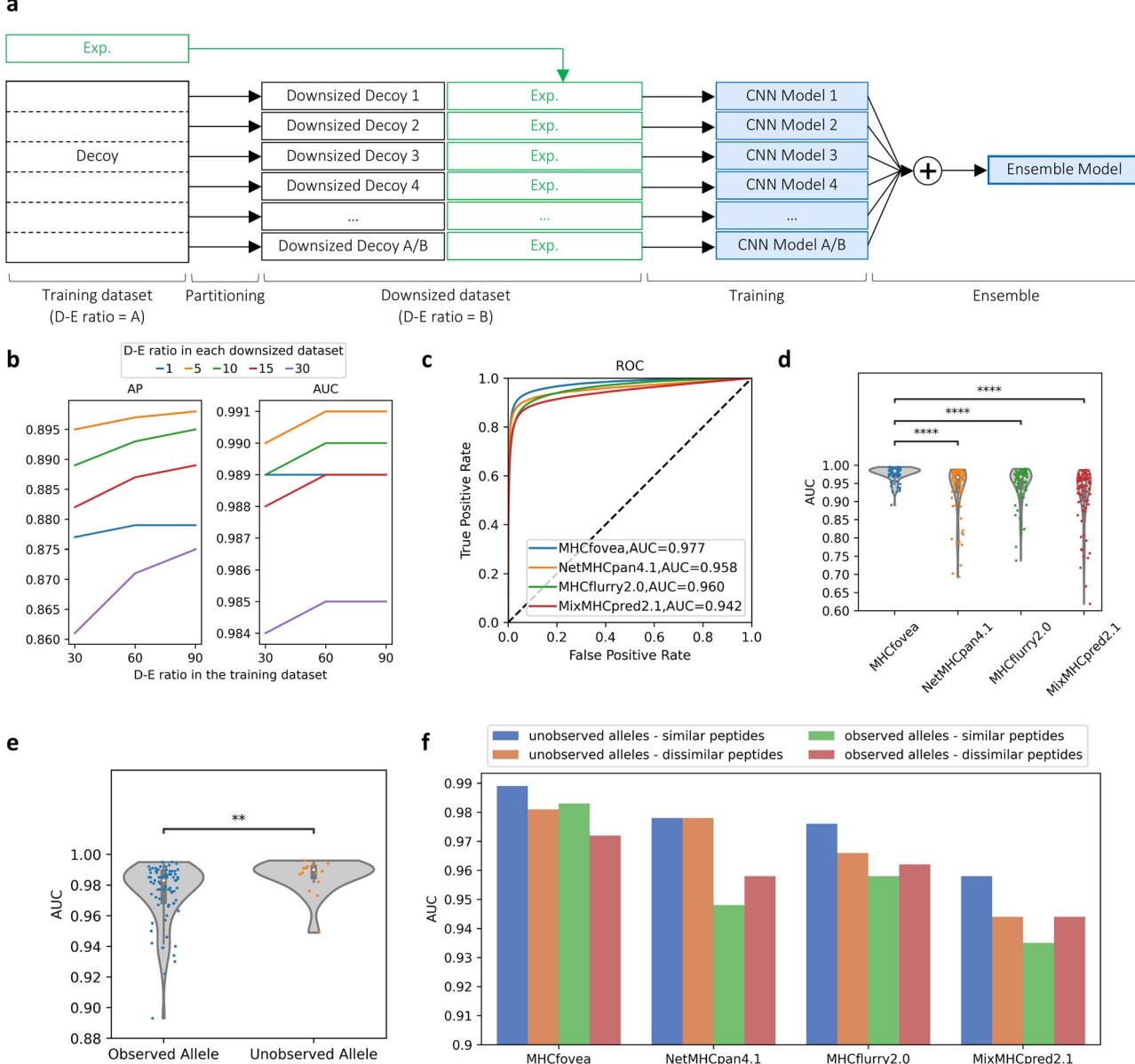

**Fig. 2 The framework and performance of the MHCfovea's predictor. a** The ensemble framework with the partitioning strategy. We first adopted the training dataset with a decoy–eluted ratio (D-E ratio) of $A$. The decoy dataset was partitioned into $A/B$ downsized decoy datasets with D-E ratio of $B$. Then $A/B$ CNN models were trained on one downsized decoy dataset along with the experimental dataset. Finally, the mean of results was calculated as the prediction score. **b** AP and AUC scores on the validation dataset of the ensemble model trained under different D-E ratios in the overall training dataset, including $A = 30$, 60, and 90, against different D-E ratios in the downsized decoy dataset, including $B = 1$, 5, 10, 15, and 30. The x-axis represents the D-E ratio in the training dataset, and the y-axis represents the metric score. Source data are provided in Supplementary Tables 2 and 3. **c–f** The following performances are all applied on the benchmark dataset. **c** The ROC curves with AUC depict the comparison between predictors. **d** The violin plot shows the distribution of AUC of each predictor by alleles ($n = 91$, one allele was removed because it is unavailable in MixMHCpred2.1). **e** Comparison of the AUC between observed ($n = 76$) and unobserved ($n = 16$) alleles. **f** The comparison of AUC on the four groups split from the benchmark dataset between predictors. In violin plots, boxplots depict the median value with a white dot, the 75th and 25th percentile upper and lower hinges, respectively, and whiskers with 1.5× interquartile ranges. $P$ values (two-tailed independent $t$ test) are shown as **$P \leq 0.01$ and ****$P \leq 0.0001$. Source data and details of the statistical analysis are provided in Supplementary Data 2, 3, and 4.

are different. It is reasonable for a machine learning task to have better performance on the groups of similar peptides than of dissimilar ones. Of note, MHCfovea still has better performance than the other predictors on the dissimilar groups.

**Selection of important MHC-I residues**. The MHC-I-binding cleft is a sequence of 182 a.a., some of which occupy highly polymorphic sites considered as decisive for epitope binding.

Therefore, we investigated the important positions using ScoreCAM[17], a kind of CAM algorithm. First, we applied ScoreCAM on positive peptides to illustrate how ScoreCAM works, since it has been widely considered that the second and last residues of peptides are anchor positions for most alleles[26]. Figure 3a (Supplementary Data 5) depicts that the anchor positions have higher mask scores than other residues, which reveals that ScoreCAM is capable of highlighting important positions in the peptide sequences.

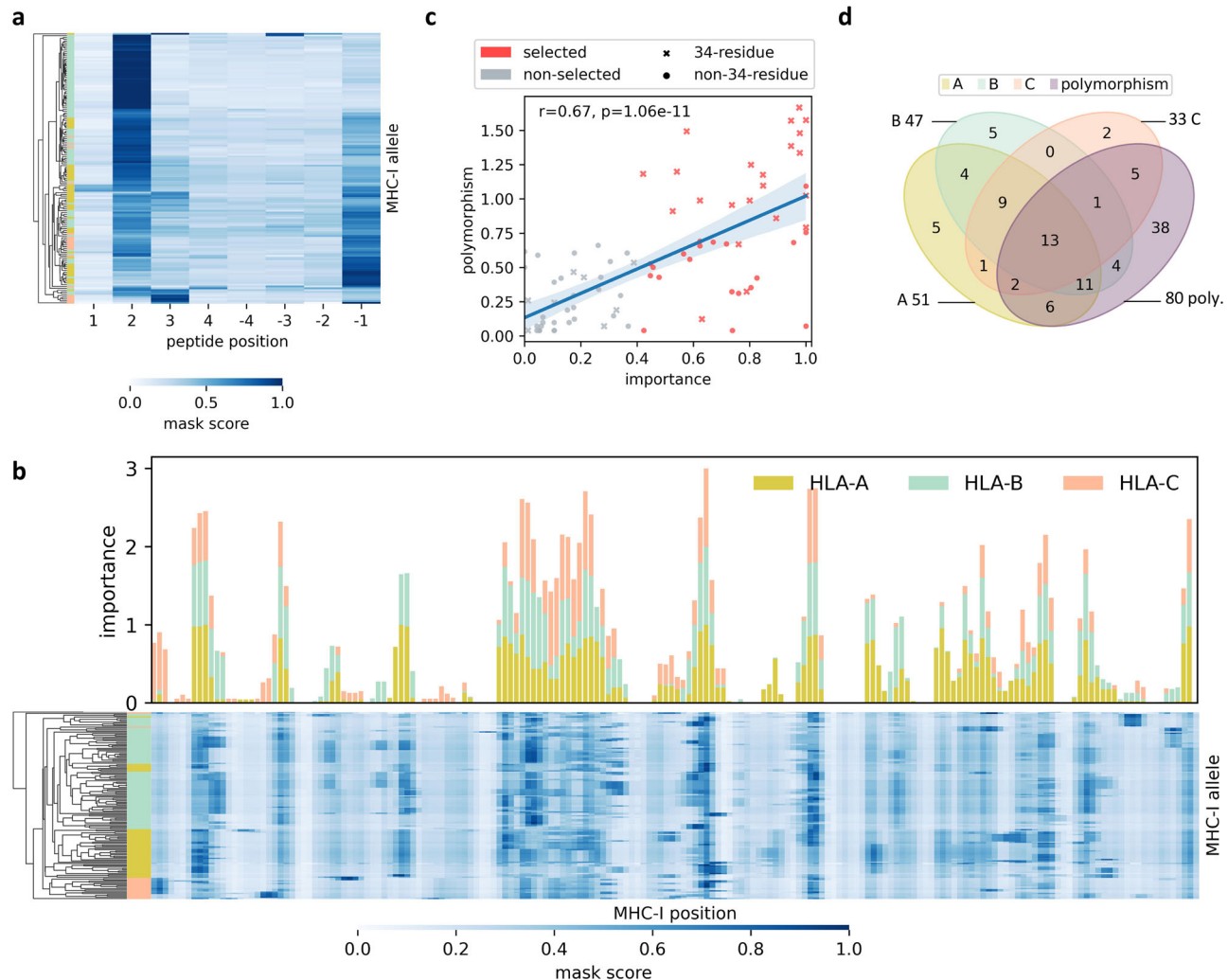

**Fig. 3 Selection of the important positions. a** A clustering heatmap of the peptide mask on each peptide position of each allele. **b** A stack plot of the position importance of HLA genes at each MHC-I residue and a heatmap of allele masks derived from ScoreCAM results with clustering on alleles. These two plots are aligned by MHC-I-binding cleft sequences, to better demonstrate the distribution of mask scores. In the stack plot, different HLA genes were counted independently due to the number of alleles with variation as well as the divergent patterns of conserved or polymorphic sequences (Supplementary Fig. 4). As for the heatmap clustering in **a**, **b**, we used Euclidean distance and unweighted average linkage for clustering mask scores, and the row color is used to label the HLA gene. **c** A scatterplot with linear correlation shows the relationship between polymorphism and importance of each polymorphic MHC-I residue ($n = 80$). Information entropy ($-\Sigma P \times \ln(P)$, where $P$ is the amino acid frequency) is used to represent the degree of polymorphism. The important positions selected using ScoreCAM are colored in red, and the 34 residues derived from NetMHCpan4.1 are cross-marked. The blue band represents the 95% confidence interval of the regression fit, and the line represents the estimated regression. **d** A Venn diagram shows the intersection of the important position set from each HLA gene and the polymorphic residue sets. Residues in the set of "$(A \cup B \cup C) \cap$ polymorphism" are selected as the 42 important positions of MHCfovea. Source data are provided in Supplementary Data 5 and 6.

Next, we focused on the positive predictions of the training dataset and obtained allele masks; briefly, every position has a mask score representing the relative importance across the 182 a.a. Figure 3b (Supplementary Data 5 and 6) shows the stack plot of importance of each HLA gene at each position and the heatmap clustering of allele masks. The importance of each position was quantified by the proportion of alleles with a mask score of >0.4. Importantly, alleles from identical HLA genes were mostly grouped together in the heatmap, consistent with the divergence of importance between different HLA genes in the stack plot. This result indicates that our model not only learned the differences between HLA-A, -B, and -C but also focused on different positions in different HLA genes.

Additionally, to evaluate the consistency of polymorphism and mask score of each position, we applied linear regression analysis on the degree of polymorphism and importance. The degree of

polymorphism was calculated by the information entropy of a.a. frequency. Owing to the divergence between HLA genes in Fig. 3b, the importance scores of HLA-A, -B, and -C were calculated separately, and the maximum one was chosen as the final importance. The activation maps derived from CAM-based approaches are not sharp enough; residues next to the real important residue could be highlighted simultaneously. This explains why some non-polymorphic positions also have high importance; therefore, before applying linear regression, we removed all non-polymorphic positions. Figure 3c (Supplementary Data 6) presents a Pearson's correlation of 0.67 ($P < 0.05$) between polymorphism and importance and reveals that highly polymorphic sites play a more important role in the predictor.

Polymorphic positions with importance >0.4 were chosen as important positions. Figure 3d presents the Venn diagram of position selection. In the end, 42 important positions were

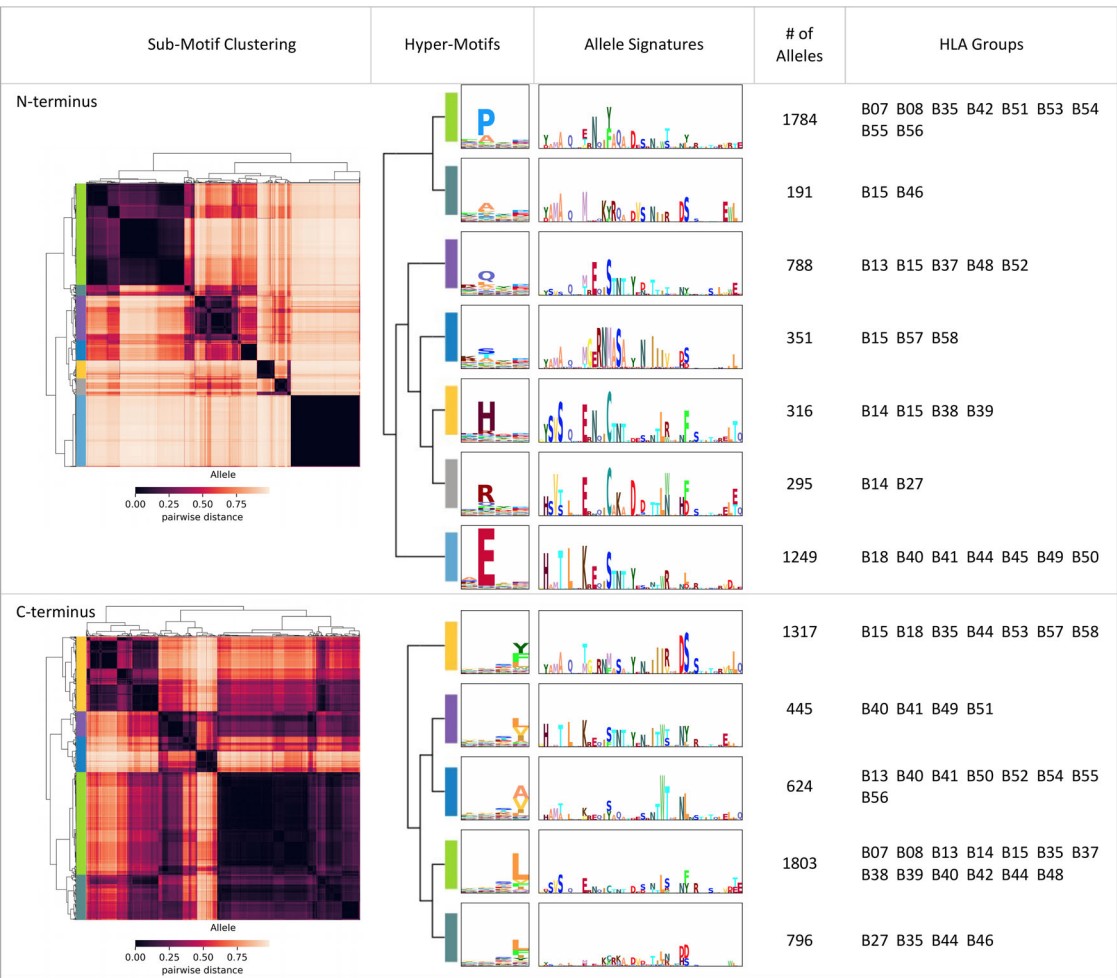

**Fig. 4 The relation between MHC-I sequences and MHC-I-binding motifs.** A summarization table of HLA-B. The MHC-I-binding motifs are divided into N- and C-terminal sub-motifs; sub-motifs are clustered by agglomerative hierarchical clustering. Hyper-motifs and the corresponding allele signatures are calculated from each sub-motif cluster. In each cluster, the number of alleles and the HLA groups with the number of alleles ≥25 are recorded in the last two columns. Source data are provided in Supplementary Data 7.

selected, and 13 of them were important in all HLA genes (Supplementary Data 6).

We compared the selected residues (42 residues) with 34 contact residues (the pseudo-sequence applied in NetMHCpan4.1)[20] in Fig. 3c. Some highly polymorphic sites are not included in the pseudo-sequence but have high importance, suggesting that some residues other than the 34 contact residues are essential for epitope binding, such as position 65 and 71.

**Expansion and summarization of MHC-I-binding motifs.** Each MHC-I allele has its own binding motif owing to the distinct MHC-I sequence. To further explore the pattern among different alleles, we computed the binding motif of alleles in the training dataset. Since the length of epitopes ranges from 8 to 15 and the important residues are usually located at the second and last positions, we focused on the first four (N-terminal) and last four (C-terminal) residues to construct an 8-a.a.-long motif for peptides bound by each allele[26]. Supplementary Fig. 5 depicts the hierarchical clustering of the binding motifs of HLA-B alleles. Some alleles, especially those of the identical HLA group (e.g., B*44), have similar binding motifs and are grouped together; however, some alleles with similar N-terminal sub-motifs have dissimilar C-terminal sub-motifs. For example, both HLA-

B*40:01 and HLA-B*41:01 have an E-dominant N-terminal sub-motif, but the former has an L-dominant C-terminal sub-motif and the latter has an A-dominant one. This motivated MHCfovea to cluster the N-terminal and C-terminal sub-motifs separately.

When exploring the relation between HLA sequences and MHC-I-binding motifs/sub-motifs, we noticed that the number of alleles in a cluster is too small to form meaningful signatures. The training dataset has only 150 alleles, a fraction of the 13,008 MHC-I alleles recorded in the IPD-IMGT/HLA database[10]; it is difficult to obtain notable MHC-I sequence patterns from such an insufficient number of alleles. Therefore, we made predictions on all available alleles to generate more binding motifs, relying on the good performance of the MHCfovea's predictor. In total, we obtained 4158 HLA-A-binding motifs, 4985 HLA-B-binding motifs, and 3865 HLA-C-binding motifs.

We then retrieved N- and C-terminal sub-motifs and clustered them into several clusters. Figure 4 (Supplementary Data 7) shows the clustering of N- and C-terminal sub-motifs of all HLA-B alleles, with 7 N-terminal and 5 C-terminal sub-motif clusters where minor clusters that have <50 alleles are neglected. For each sub-motif cluster, we calculated the hyper-motif and the corresponding allele signature to represent the preference of binding motifs and a.a. at the important positions (Fig. 4, Supplementary Figs. 6 and 7, and Supplementary Data 7). Of note, in each cluster, 50 alleles from each HLA group were

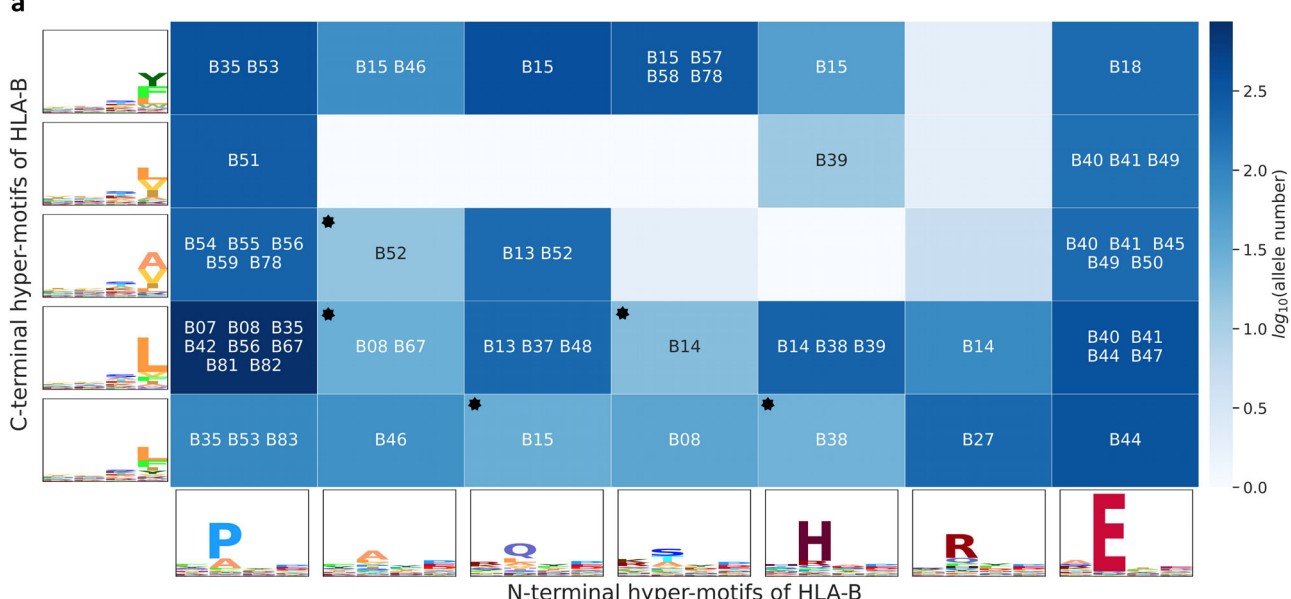

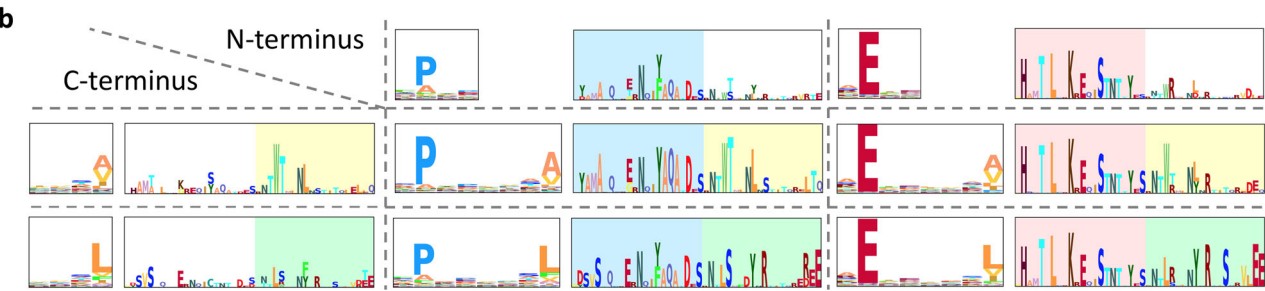

**Fig. 5 The combination map of N- and C-terminal hyper-motifs. a** The binding motif of an allele is a combination of an N-terminal and a C-terminal hyper-motif. After allocating all the alleles into the combination map, the cell color is determined by log₁₀(number of alleles in the cell). In each cell with an allele number >10, the maximal HLA group and HLA groups with an allele number ≥25 or with a proportion (the allele number in the cell to the overall number of an allele group) >0.1 are listed. **b** The relation of a combination to its hyper-motifs. Four combinations are used as an example to illustrate the consistent signatures across different cells in the same column or row. The header column and header row consist of two N-terminal and two C-terminal clusters, respectively. Then, alleles of a cell, the combination of the N-terminal (column) and C-terminal (row) clusters, are used to generate the corresponding hyper-motif and allele signature. The color boxes are used to highlight the similar part of allele signatures. Source data are provided in Supplementary Data 8.

randomly sampled to construct the allele signature to reduce the imbalance between different HLA groups. Notably, the pattern of binding motifs and allele signatures are partly interpretable with the property of a.a. In Fig. 4, the first cluster of C-terminal hyper-motifs is composed of aromatic residues (e.g., Y and F), whereas the second and third clusters are composed of aliphatic a.a. (e.g., L, V, I, and A). Moreover, the fifth and sixth clusters of N-terminal hyper-motifs dominated by basic a.a. (H and R) with similar allele signatures, indicating that MHC-I–peptide binding depends on physicochemical properties to some extent.

To investigate the distribution of allele groups with respect to the combinations of N- and C-terminal clusters, we plotted the combination heatmap in Fig. 5a (Supplementary Fig. 8 for HLA-A and -C and Supplementary Data 8), which in total has 35 combinations (7 N-terminus × 5 C-terminus) for HLA-B. Interestingly, five unobserved combinations, not present in the training dataset, were discovered by MHCfovea via the pattern learned from the observed combinations. In Fig. 5b, we presented four combinations of N- and C-terminal clusters. The noticeable residues of N- and C-terminal hyper-motifs are mostly located in the first half and last half part of allele signatures, respectively,

which is consistent with the binding structure of MHC-I molecules[27]. For example, the E-dominant cluster has noticeable residues in the first half part of the allele signature; these residues are highly conserved in not only different combinations but also the cluster, which enhances confidence of the key residues highlighted in the allele signature.

**Disclosure of the HLA groups falling into multiple sub-motif clusters.** Overall, alleles within the same HLA group were clustered into the same sub-motif cluster. However, Fig. 4 shows that some HLA groups, such as B*15 and B*56, fell into multiple sub-motif clusters. An HLA group is defined as a multi-cluster HLA group if its alleles fall into multiple clusters and the second large cluster contains the number of alleles ≥25 or the ratio to the total allele number of this group >0.1; MHCfovea identified 27 multi-cluster HLA groups, listed in Supplementary Table 8.

Here we used the important positions and expanded alleles to further investigate the multi-cluster HLA groups. Figure 6a (Supplementary Data 9) shows that the difference in polymorphism between multi-cluster and mono-cluster HLA groups is

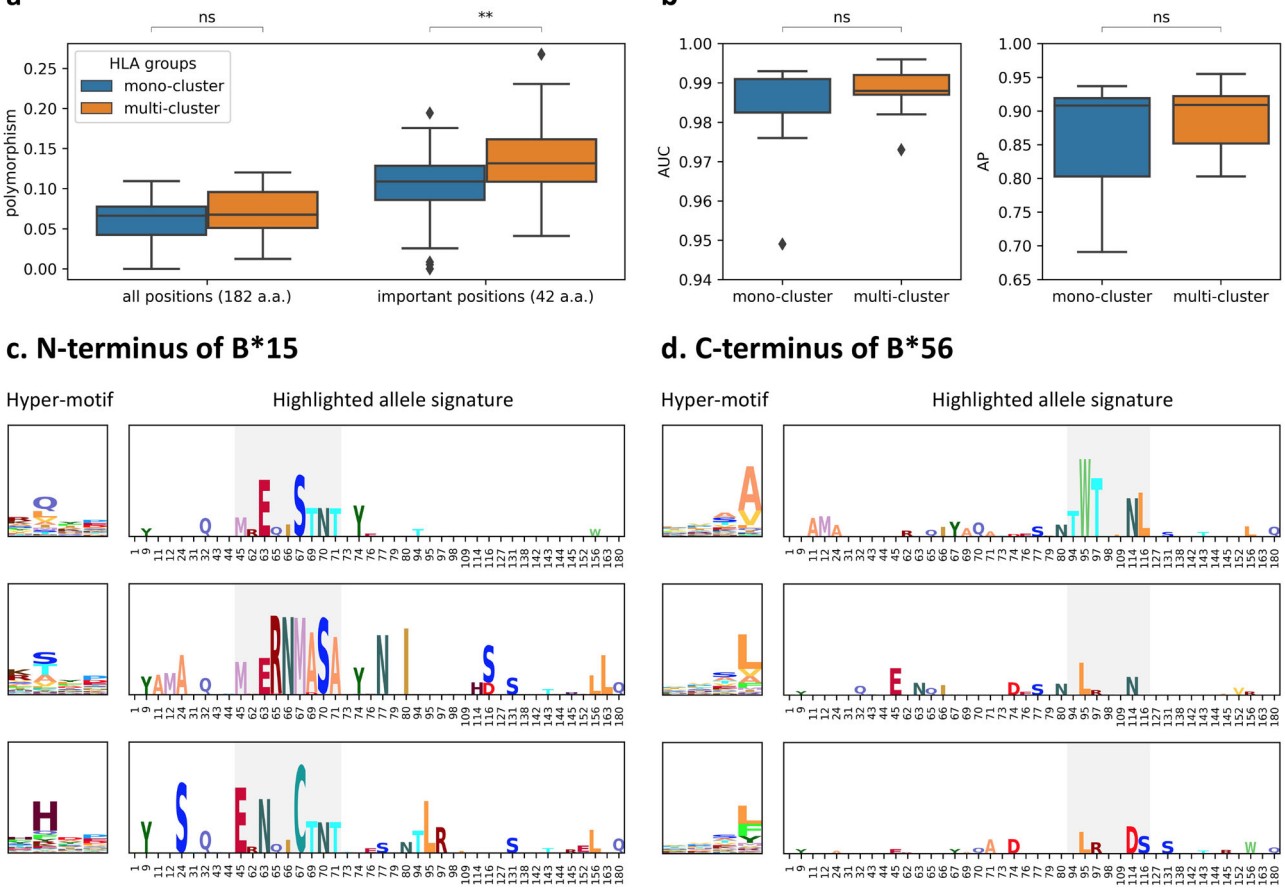

**Fig. 6 Characteristics of the HLA groups falling into multiple sub-motif clusters. a** Polymorphism on all 182 amino acids or the important positions of mono-cluster (group number = 44) or multi-cluster (group number = 27) HLA-groups. **b** AUC and AP of unobserved alleles grouped by mono-cluster (allele number = 7) or multi-cluster (allele number = 9) HLA-groups. **c, d** The hyper-motifs and highlighted allele signatures of the N-terminal sub-motif clusters of B*15 (**c**) and the C-terminal sub-motif clusters of B*56 (**d**). The box colored in gray is used to highlight the polymorphic sites. Boxplots depict the median value with a middle line, the 75th and 25th percentile upper and lower hinges, respectively, whiskers with 1.5× interquartile ranges, and points as outliers. *P* values (two-tailed independent *t* test) are shown as "ns" no significance and **P ≤ 0.01. Source data and details of the statistical analysis are provided in Supplementary Data 9 and 10.

significant considering the important positions, but not all 182 a.a. Figure 6b (Supplementary Data 10) shows that MHCfovea has good performance with respect to unobserved alleles for both the mono- and multi-cluster HLA groups. Figure 6c, d demonstrate hyper-motifs and highlighted allele signatures of multi-cluster HLA groups. Figure 6c shows three major N-terminal sub-motif clusters of B*15; the gray box highlights the highly polymorphic sites, especially position 67, which may contribute to different MHC-I-binding motifs. Additionally, position 65 and 71, not selected in the pseudo-sequence of NetMHCpan4.1 (Fig. 3c), are highlighted in the second cluster of Fig. 6c, supporting that some important positions beyond 34 contact residues are also decisive for the binding motif. On the other hand, Fig. 6d shows three major C-terminal sub-motif clusters of B*56; in the B*56 HLA group, only B*56:01 was present in the training dataset, which reveals that another two clusters were discovered by MHCfovea after allele expansion. In summary, these results demonstrate some notable patterns of MHC-I sequences beyond HLA groups, corresponding to some specific sub-motifs.

## Discussion
Antigen discovery is composed of two major steps, antigen presentation and T cell recognition[1]; several researches have built accurate predictors for antigen presentation, especially

MHC–peptide binding[12]. However, the decisive residues of MHC sequences for peptide binding are still unspecified. A few studies have explored the pattern of MHC sequences and peptides[14–16]; nevertheless, owing to the limited number of alleles with experimental measurements, it is hard to conclude the relation of MHC sequences and binding motifs from all MHC alleles.

Here we developed MHCfovea for predicting binding probability and providing the connection between MHC-I sequences and binding motifs. MHCfovea's predictor outperformed the other predictors via an ensemble framework with downsampling to solve the data imbalance between decoy and eluted peptides. To focus on the important positions determining the binding motifs, MHCfovea selected 42 a.a. of MHC-I sequences based on 150 observed alleles using ScoreCAM. After expanding the knowledge from observed alleles to unobserved alleles (total number: 13,008), MHCfovea delivered 32 pairs (HLA-A: 13, HLA-B: 12, and HLA-C: 7) of hyper-motifs and allele signatures on 42 important positions to reveal the relation of MHC-I sequences and binding motifs. In addition, MHCfovea discovered some unobserved combinations of N- and C-terminal sub-motifs with the support from high similarity between allele signatures. Finally, MHCfovea disclosed some multi-cluster HLA groups, such as B*15 and B*56, and highlighted the key residues to determine the different binding motifs.

Since the positive allele–peptide pairs in the benchmark data have a high ratio of peptides (32.8%) that were present in the training data, it is not clear if the good performance of MHCfovea came from the memorization of the peptides in the positive pairs of the training data. To clarify this point, we built an artificial dataset by pairing all the alleles in the benchmark dataset and the positive peptides in the training dataset. Supplementary Fig. 9 depicts the distribution of the artificial dataset, which is close to the negative data in the benchmark. In other words, the artificial pairs are recognized as negative samples mostly in the MHCfovea's predictor. This result indicated that the MHCfovea's predictor actually recognized the binding peptides via the sequence patterns rather than memorizing all the positive peptides in the training data.

Some limitations of MHCfovea are addressed here. First, the unobserved binding motifs are derived from predictions. Although MHCfovea has an accurate performance in the context of unobserved alleles, the total number of alleles with experimental data is a small fraction of available MHC-I alleles. Second, sub-motifs with a dominant a.a. can be clustered notably. In contrast, sub-motifs of HLA-C mostly with no dominant a.a. have neither obvious clusters nor indistinguishable allele signatures; therefore, it is difficult to determine the relation between binding motifs and MHC-I sequences on such alleles. Additionally, the number of clusters is fixed once summarization is completed. In this study, some minor clusters with <50 alleles were neglected, and in the end 32 major clusters are presented in our summarization. Most alleles (12,919 in 13,008, 99%) belong to one N-terminal and one C-terminal cluster within these 32 clusters. If new alleles are appended in the future, the process of allele extension and summarization can be reperformed to generate a new set of clusters.

As for the binding prediction, the testing dataset is the same for each predictor, but the training dataset is not. Although MHCfovea has no advantage on the numbers of alleles and peptides when compared with NetMHCpan4.1 or MHCflurry2.0 (Supplementary Table 4 and Supplementary Fig. 2), the lack of a public training dataset is still a limitation for comparison between different algorithms. Furthermore, MHCfovea is only trained on mono-allelic measurements; adding multi-allelic data to the training dataset increases not only the number of peptides but also the diversity of MHC-I alleles. Alvarez et al.[28] designed a semi-supervised method to associate each ligand to its MHC-I allele, which can potentially deal with the ambiguous annotation on multi-allelic data. In the future, we will incorporate this method with MHCfovea to enlarge the number of observed alleles; we anticipate increasing the number of experimental data can further improve model performance and the quality of the summarization of MHCfovea. Furthermore, a complete immune response depends on the recognition of MHC-I–peptide complexes by T cells. Building a model for T cell immunogenicity following MHCfovea is expected to promote the contribution of computational approaches on antigen discovery.

In summary, MHCfovea successfully connects MHC-I alleles with binding motifs via deep learning. MHCfovea's predictor expanded the knowledge of MHC-I-binding motifs from 150 alleles to 13,008, which were further summarized into pairs of hyper-motifs and allele signatures. The large number of allele sequences realized the generalization of allele signatures connected to distinct binding motifs correspondingly. Antigen discovery and vaccine design can be facilitated by knowing such clustered alleles and their key residues. Additionally, MHCfovea reveals some multi-cluster HLA groups, which provided additional examination for allele similarity beyond the allele group, based on the 42 important positions of MHC-I uncovered by MHCfovea.

## Methods

**Preparation of MHC-I sequences**. We used the IPD-IMGT/HLA database (version 3.41.0)[10] as a reference for MHC-I sequences and used peptide-binding clefts annotated in the UniProt database[29] as the target binding region. Of note, the peptide-binding cleft, composed of α-1 and α-2 regions, is a protein sequence with 182 a.a. and is critical for epitope presentation[9]. We used the alignment file from the IPD-IMGT/HLA database and obtained the corresponding sequences to build a peptide-binding domain database of all MHC-I alleles for the development of the proposed pan-allele-binding predictor adopted by MHCfovea.

**Preparation of peptide data**. Experimental data of binding and ligand elution assays, especially mass spectrometry (MS), were collected from Immune Epitope Database and Analysis Resource (IEDB)[30], the most comprehensive immunopeptidome database. Because MHCfovea is a binary classifier for MHC-I–peptide binding, all measurements were labeled with 0 and 1. For the binding assays, an $IC_{50}$ of 500 nM was set as the upper bound for the positive label. As for ligand elution assay, all samples were labeled as positive.

The binding assay dataset generated in 2013 was directly downloaded from IEDB. To focus on the prediction of four-digit human MHC-I alleles (for example, A*01:01), non-human, mutant, and digital-insufficient MHC-I alleles were excluded. The peptides were restricted to 8–15-mers and this setting covered most epitopes[19]. The MS dataset was exported from IEDB on 2020/07/01; the following filters were used: linear epitopes, human species, MHC class I, and positive MHC ligand assay. Both 4-digit human alleles and peptides with a length of 8–15 a.a. were selected, following the same selection strategy as above. After filtration, the dataset consisted of 515,110 measurements across 150 alleles.

**Separation of the training, validation, and benchmark datasets**. To build an isolated testing benchmark, we considered a single experimental reference selected from the previous ligand elution assay dataset. The MHC-I immunopeptidome built by Sarkizova et al.[15] is the largest mono-allelic MS dataset, comprising 127,371 measurements across 92 alleles and was, therefore, chosen as the testing benchmark in this study. The binding assay dataset and the MS dataset excluding the experimental data used in the benchmark were combined to build the training dataset (95%) and the validation dataset (5%). In addition, to avoid duplication between training and benchmark datasets, we excluded peptides with identical allele and peptide sequences from the training and validation datasets and retained them in the benchmark dataset.

**Preparation of decoy peptides**. As the MS data only provide positive results, we prepared a decoy dataset to be used as negative results. We created two types of decoy peptides, "protein decoy" and "random decoy," both extracted from the UniProt proteome. "Protein decoy" refers to the peptides that were generated from the same protein as an eluted peptide, whereas "random decoy" refers to the peptides that were randomly extracted from the UniProt proteome. For each eluted peptide in the benchmark and validation datasets, we created two protein decoy peptides and two random decoy peptides for each length of 8–15 (a.a.). Duplicated peptides with identical allele and peptide sequence were excluded. In the end, both benchmark and validation datasets had a D-E ratio of 30, which is close to that of the dataset in NetMHCpan4.1[20]. On the other hand, in the training dataset, to evaluate the effect of the D-E ratio on model performance, we generated decoy peptides with D-E ratios >30. For each eluted peptide, we created two protein decoy peptides and ten random decoy peptides for each length of 8–15 (a.a.). We only enlarged the number of random decoy peptides, because it was difficult to select more different unique peptides from a single protein (protein decoy peptides) with a short length. In the end, the training dataset had a D-E ratio of 90, which is three times that in the validation and benchmark datasets. The number of data instances in each dataset is listed in Supplementary Table 1, and the data number by alleles of the training, validation, and benchmark datasets is recorded in Supplementary Data 1.

**CNN model architecture**. The predictor adopted by MHCfovea is an ensemble model of multiple CNN model. A CNN model takes the allele (182 a.a.) and peptide (8–15 a.a.) sequences as input; both sequences are encoded with a one-hot encoder of a.a. The CNN model architecture is shown in Supplementary Fig. 1. Before concatenation, the encoded vectors are passed through several convolution blocks separately. The convolution block is composed of a 1D convolution layer with kernel size 3, stride 1, and zero-padding 1; a batch normalization layer; a ReLU activation layer; as well as a max-pooling layer. In the allele part, sequences are passed through four convolution blocks and downsized to a 15-long matrix. In the peptide part, all sequences are padded with "X" as an unknown a.a. to 15-long at the end of the sequence, the maximal length of peptides, and three convolution blocks are applied. After concatenation in the dimension of filters, the matrix is passed through another two convolution blocks with the replacement of the last max-pooling layer by a global max-pooling layer followed by a fully connected layer and a sigmoid operator. Finally, a prediction score is obtained to represent the binding probability of MHC-I and peptide sequences.

**Model training.** MHCfovea uses binary cross entropy as its loss function and the Adam optimization algorithm with the weight decay of $10^{-4}$ as the optimizer. The number of training epochs was set to 30, and the best model state was chosen after epoch 24 via the loss of the validation dataset to avoid overfitting. The hyperparameters, including the batch size and learning rate, were selected via the grid search optimizer based on AP of the validation dataset. The batch size of 32 from the options [16, 32, 64] and the learning rate of $10^{-4}$ from the options [$10^{-5}$, $10^{-4}$, $10^{-3}$] were selected (Supplementary Table 9). In addition, the learning rate scheduler was used to adjust the learning rate during the training process. Of note, the learning rate was reduced to $10^{-5}$ after epoch 15 and to $10^{-6}$ after epoch 24.

**Performance metrics.** We used four metrics, AUC, AUC0.1, AP, and PPV, to evaluate the performance of our model as well as that of other predictors. The AUC is a curve of the true positive rate (TPR) against the false positive rate (FPR). AUC0.1 has a restriction of the FPR under 0.1. AP is the area under the precision-recall curve created by plotting the precision against TPR, also called recall. PPV (positive predictive value) is defined as Eq. (1), where $N$ is the number of positive measurements.

$$\text{PPV} = \frac{\text{positive predictions within top } N \text{ predictions}}{N} \quad (1)$$

In addition, we calculated these metrics in the context of every allele to evaluate the distribution of allelic performance.

**Threshold and %rank for the prediction score.** To explain prediction score more explicitly, a positive threshold and the %rank score, the percentage of ranking among background peptides, were provided. The positive threshold was set according to the maximal F1 score on the validation dataset. As for the %rank, 10,000 random peptides extracted from the UniProt database were built as the background peptides to calculate the %rank of each prediction score. When a prediction receives a %rank of 0.5, it means a peptide binds to MHC-I more probably than 99.5% random peptides.

**Comparison with other predictors.** NetMHCpan4.1[20], MHCflurry2.0[25], and MixMHCpred2.1[16], well-known MHC-I–peptide binding predictors, were compared with the MHCfovea's predictor. For MHCflurry2.0, we used the variant model of MHCflurry2.0-BA, the only one trained without our benchmark dataset. Both NetMHCpan4.1 and MHCflurry2.0 are compatible with all kinds of a.a. and 8–15 length peptides; however, for MHCflurry2.0, we had to replace a.a. beyond 20 human-required a.a. with "X" as an unknown a.a. On the other hand, MixMHCpred2.1 only allows 8–14 length peptides and sequences within 20 a.a.; therefore, for accurate comparison, we removed peptides with other a.a. or those >14 a.a.

First, we tested all models directly on the benchmark dataset and calculated performance metrics for comparison. The output of MHCflurry2.0 was the $IC_{50}$ of the binding affinity; therefore, we used a function $(1 - \log_{50,000}(x))$ to transform the binding affinity into binding probability. Then the performances of these models were tested by allele to evaluate the confidence between different alleles. In total, there are two types of results because of the peptide availability of MixMHCpred2.1 depicted in Supplementary Data 2 and 4.

**Class activation mapping.** We applied CAM on our model for interpretation purposes. CAM-based approaches provide the explanation for a single input with activation maps from a convolution layer. There are several CAM-based methods, including CAM[18], GradCAM[31], GradCAM++[32], and ScoreCAM[17]. ScoreCAM was chosen due to its stability on the former convolution layer. We applied ScoreCAM on the second convolution block before the max-pooling layer of the MHC part (Supplementary Fig. 1). We focused on positive predictions with prediction scores >0.9. The mean of ScoreCAM scores from positive predictions of a single allele was calculated as the final result called "epitope mask" for epitope part and "allele mask" for allele part used in Fig. 3. Of note, in epitope and allele masks, every position has a score representing the relative importance across the 8-a.a.-long and 182-a.a.-long sequences, respectively.

**Selection of the important positions.** The training dataset with 150 alleles composed of 46 HLA-A, 85 HLA-B, as well as 19 HLA-C alleles was used to select the important positions, and both allele masks and a.a. polymorphism were taken into consideration. Owing to the divergence between HLA genes, positions from different HLA genes were chosen separately. First, we calculated the importance of each position for each HLA gene. The importance of a position was quantified as the proportion of alleles with mask scores >0.4 (set heuristically). Then, for each HLA gene, residues with importance >0.4 (also set heuristically) were selected; however, those with no polymorphism (all alleles had the same a.a.) were dropped. Then we combined the selected positions from each HLA gene as important positions of our model. In total, we selected 42 important positions, including positions 1, 9, 11, 12, 24, 31, 32, 43, 44, 45, 62, 63, 65, 66, 67, 69, 70, 71, 73, 74, 76, 77, 79, 80, 94, 95, 97, 98, 109, 114, 116, 127, 131, 138, 142, 143, 144, 145, 152, 156, 163, and 180 of the MHC-I peptide-binding cleft sequence (182 a.a.).

**Prediction of all alleles.** With the good performance of MHCfovea's predictor on unobserved alleles, we predicted the binding probability of each allele against 254,742 peptides (including all ligand elution data and some decoy peptides whose number was the same as ligand elution data of the benchmark dataset). In total, 3.3 billion pairs of peptide–allele were tested. The peptides with a prediction score >0.9 (~78 million peptides) were sent to the summarization module to calculate the binding motif for each allele. Each MHC-I-binding motif with 8 a.a. was composed of the first four residues (N-terminal) and the last four residues (C-terminal). In total, we obtained 4158 HLA-A-binding motifs, 4985 HLA-B-binding motifs, and 3865 HLA-C-binding motifs in the summarization step.

**Sequence motifs.** The sequence motif is the pattern of a set sequences. There are some types of matrices, including position probability matrix (PPM), position weight matrix, and information content matrix (ICM), used to represent the sequence motif. In this study, we used PPM to calculate the MHC-I sequence motif and ICM to calculate the MHC-I-binding motif. From a set $S$ of $M$ aligned sequences of length $L$, the elements of the PPM are calculated from Eq. (2), where $I$ is an indicator function.

$$\text{PPM}_{i,j} = \frac{1}{M}\sum_{k=1}^{M} I(S_{k,j} = i), \quad \begin{array}{l} i = \{20 \text{ amino acids}\} \\ j = 1, ..., L \end{array} \quad (2)$$

The ICM is used to correct PPM with background frequencies and highlight more important residues. The elements of the ICM are calculated from Eq. (3), where the background frequency $B$ is 0.05 (=1/20) for each a.a.

$$\text{ICM}_{i,j} = \text{PPM}_{i,j} \sum_{m \in \{20 \text{ amino acids}\}} \left[ \text{PPM}_{m,j} \times \frac{\log_2\left(\text{PPM}_{m,j}\right)}{B} \right] \quad (3)$$

**Sub-motif clustering.** An MHC-I-binding motif with 8 a.a. was split into an N-terminal sub-motif with the first 4 residues of the binding motif and a C-terminal sub-motif with the last 4 residues of the binding motif. Consequently, a sub-motif is represented by a $4 \times 20$ (the number of a.a.) ICM. Before clustering, the pairwise distance of each sub-motif was calculated via cosine metric. Then we used agglomerative hierarchical clustering with cosine metrics and maximum linkage to cluster the pairwise distance. Different numbers of clusters were set for different HLA genes and termini manually, and minor clusters with <50 alleles were neglected.

**Hyper-motifs and allele signatures.** Hyper-motifs and allele signatures are both used to demonstrate the characteristics of a specific group of alleles. Hyper-motifs representing the MHC-I-binding motif of alleles were calculated from the element-wise mean of motif or sub-motif matrices. Allele signatures disclose the preference of a.a. at important positions. For each sub-motif cluster, we sampled 50 alleles from each HLA group on a two-digit level to balance the allele number of each group because of two reasons. First, alleles with the same HLA group have similar MHC-I sequences, which may lead to similar binding motifs (Supplementary Fig. 5). Second, there is a huge variation among the allele number of different HLA groups. For example, HLA-B*07 has 394 alleles, but HLA-B*56 only has 69 alleles. Of note, if the allele number of an HLA group was <50, all alleles were selected.

Afterward, to generate the allele signature matrix (ASM), we had to calculate a background PPM ($\text{PPM}^{\text{background}}$) from all sampled alleles of an HLA gene and a PPM of sampled alleles from a specific sub-motif cluster ($\text{PPM}^{\text{cluster}}$). On the other hand, to calculate the sequence pattern of HLA groups, we also calculated the PPM of alleles from an HLA group in a specific sub-motif cluster ($\text{PPM}^{\text{group}}$). The $\text{ASM}^{\text{cluster}}$ was defined as the positive part of the difference between $\text{PPM}^{\text{cluster}}$ and $\text{PPM}^{\text{background}}$ in Eq. (4); the $\text{ASM}^{\text{group}}$ was defined as the difference between $\text{PPM}^{\text{group}}$ and $\text{PPM}^{\text{background}}$ in Eq. (5), where Iverson bracket was used to set positive elements and the others as 1 and 0, respectively.

$$\text{ASM}^{\text{cluster}} = f^+\left(\text{PPM}^{\text{cluster}} - \text{PPM}^{\text{background}}\right), \quad f^+(x) = \max\left(f(x), 0\right) \quad (4)$$

$$\text{ASM}^{\text{group}} = g\left(\text{PPM}^{\text{group}} - \text{PPM}^{\text{background}}\right), \quad g(x) = [x > 0] \quad (5)$$

For instance, we used 1790 sampled HLA-B alleles to generate the $\text{PPM}^{\text{background}}$ of HLA-B and 502 sampled alleles of the P-dominant sub-motif cluster (Fig. 4) to produce the $\text{PPM}^{\text{cluster}}$. The allele signature of the P-dominant N-terminal sub-motif in the header row of Fig. 4 was calculated from the difference of these two probability matrices.

In addition, the highlighted allele signature demonstrated in Fig. 6c, d was used to highlight the similarity of allele signatures of the specific alleles and of the corresponding sub-motif clusters. We implemented the element-wise product to get the highlighted allele signatures in Eq. (6), where $L$ is the sequence length and HASM is the matrix of the highlighted allele signature.

$$\text{HASM}_{i,j} = \left(\text{ASM}^{\text{group}} \circ \text{ASM}^{\text{cluster}}\right)_{i,j}, \quad \begin{array}{l} i = \{20 \text{ amino acids}\} \\ j = 1, ..., L \end{array} \quad (6)$$

**Statistics and reproducibility.** For all comparisons of performance, we used two-tailed independent $t$ test and set the criterion for statistical significance as $P < 0.05$.

For the relation between polymorphism and importance of each polymorphic MHC-I residue, we fit a linear regression along with 95% confidence interval (Fig. 3c).

**Reporting summary**. Further information on research design is available in the Nature Research Reporting Summary linked to this article.

## Data availability

Several public databases were used in this study, including Immune Epitope Database and Analysis Resource (IEDB) (https://www.iedb.org/) for experimental measurements, UniProt (https://ftp.uniprot.org/pub/databases/uniprot/current_release/knowledgebase/complete/uniprot_sprot.fasta.gz) for decoy peptides, and IPD-IMGT/HLA (https://github.com/ANHIG/IMGTHLA/tree/3410) for MHC-I allele sequences. Research data files supporting this study, including the peptide-binding cleft sequence of MHC-I alleles; the training, validation, and benchmark datasets; the prediction of the validation and benchmark datasets; and the prediction of the allele expansion are available from Mendeley Data (https://doi.org/10.17632/c249p8gdzd.3)[33]. Source data for all figures are provided in Supplementary Data.

## Code availability

The source code of the research and the MHCfovea's predictor are freely available at GitHub (https://github.com/kohanlee1995/MHCfovea) and Mendeley Data[33] for academic non-commercial research purposes. All source codes are based on Python (v3.6.9) and its packages, including numpy (v1.18.2), pandas (v1.0.3), scikit-learn (v0.22.2), pytorch (v1.4.0), matplotlib (v3.2.1), seaborn (v0.10.0), logomaker (v0.8). Numpy, pandas, and scikit-learn, are used for data analysis; pytorch is used for deep learning; matplotlib, seaborn, and logomaker are used for visualization. The website for the summarization of MHCfovea is available at https://mhcfovea.ailabs.tw.

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

## Acknowledgements

We thank Tsung-Ting Hsieh and Hung-Ching Chang from Taiwan AI Labs for providing computational and mathematical advices. We acknowledge support from the Ministry of Science and Technology, Taiwan (MOST 109-2221-e-002-161-MY3).

## Author contributions

K.-H.L., Y.-C.C. and C.-Y.C. designed the study. K.-H.L. prepared and analyzed the data; developed, validated, and interpreted the predictor; summarized the predicted results; and wrote the manuscript. Y.-C.C. designed the figures of the MHCfovea overview and CNN model framework. T.-F.C. built the website. H.-F.J., H.-K.T. and C.-Y.C. revised the manuscript.

## Competing interests

The authors declare no competing interests.
