## [Peer Review File · Communications Biology]

Peer Review Information

Manuscript title: Connecting MHC-I-binding motifs with HLA alleles via deep learning

Reviewer comments & decisions:

Reviewer comments, first version:
--

REVIEWER COMMENTS

Reviewer #1 (Remarks to the Author: Overall significance):

Lee et al. present a novel method, MHCfovea, for predicting peptide-binding by MHC molecules and inferring peptide motifs and MHC allele signatures using deep learning.

MHCfovea addressed the imbalanced classification of MHC-peptide binding prediction by taking advantage of the ensemble strategy with down-sampling. In the evaluation with independent data of eluted ligand, MHCfovea outperformed other methods. MHCfovea detected important positions in MHC sequences using ScoreCAM. They detected multiple binding motifs, and inferred meaningful hyper-motifs with allelic signatures by clustering binding motifs.

The manuscript is well written and convincing the value of their method. However, there are several points that should be addressed.

Major points:

- The improvement of prediction accuracy in prediction of peptide-binding is valuable. Could you describe more about whether the improvement is due to just the difference in a dataset used in training or any advantages in the algorithm? Which previous methods were trained on the current data set? If the improvement is probably due to just the difference in datasets, it should be noted as such.
- The idea of outputting probability scores is good; however, I wonder if there is a downside to not outputting %rank, which is commonly used in other tools. What is the practical threshold for the score that shows significant bindings, and is it validated?
- It is interesting to evaluate the relationship between the degree of polymorphism and the sensitivity

maps derived from ScoreCAM. However, I am not familiar with ScoreCAM, but I slightly wonder if the gradient is likely to be large where many categorical variables can be taken due to the original nature of the algorithm.

- The method of tuning hyper-parameters of the deep learning model should be written. Which data were used and what optimization method was used?

Minor points:

- In page 8, the first sentence of the third paragraph of "Selection of important MHC-I residues", I cannot understand why "non-polymorphic" positions were chosen as important positions. Isn't it "polymorphic"?

- In page 9 and 10, the third sentence of the first paragraph and the fourth sentence of the third paragraph, the knowledge that the second and last positions of residues are important might be general; however, should be supported by citing a literature.

- Some of the figures may be difficult to understand because of missing labels and legends (e.g. horizontal axis of Fig. 3a, dot legend of Fig. 3b (a round dot is all residues?), and axis and heatmap label of Fig 5a).

- Some abbreviation should be decoded the first time they are mentioned in the main text (e.g. AUC, PR).

Reviewer #1 (Remarks to the Author: Impact):

The implementation of the algorithm is good, but the idea may not be fully novel. A study of predicting peptide-binding of MHC molecules using deep learning is now commonplace

Reviewer #1 (Remarks to the Author: Strength of the claims):

The novelty in this work seems to be (1) the improvement in accuracy probably due to the ensemble learning; (2) the detection of important positions of MHC molecules visualization of the trained model, and binding-motif exploration

Reviewer #1 (Remarks to the Author: Reproducibility):

To strengthen reproducibility, in performance comparisons with the previous methods, using the same input data would be warranted.

Reviewer #2 (Remarks to the Author: Overall significance):

This work introduces MHCfovea, a deep Neural Network (NN) to predict the interaction between couples of MHC-I alleles and peptides. Along the prediction process, MHCfovea also shows which amino acids in the MHC-I target are deemed of interest in the interaction. This information can be used to build allele- and peptide-specific signatures through unsupervised learning (clustering).

The text clearly states the objectives of this study and is overall well written. The code, available on github, is well organised and easy to use for evaluation (the training code is not available). The performance of the predictor seems to exceed the state of the art.

Reviewer #2 (Remarks to the Author: Impact):

Although this is not the first NN-based MHC-I binding predictor, it seems to me like it is the first to include elements from deep learning frameworks that can be used to explain the predictor's choices and learn new information about the underlying biological process (i.e. which amino acids play a role in the binding). This seems like a good idea, especially in a moment where explainability in AI is very much an issue.

Reviewer #2 (Remarks to the Author: Strength of the claims):

However, I have to point out some important flaws in the methods of this study that undermine the credibility of its results and have to be addressed before the study can be published.

1) The main issue is the way the train, validation and benchmark sets are built. Although the authors state in the text that no allele/peptide couple in the benchmark set is identical to any in the train/validation sets, I am not convinced that this is enough to avoid information leakage between the sets and, subsequently, to an overestimation of the performance of the predictor. Each NN in the ensemble is made of approximately 500k weights, while the training set is made of approximately of 260k positive (binding) samples. A NN of such size could easily overfit the positive data so that these samples are "memorized" in it, which would make it unreliable when presented with previously unseen alleles or peptides. Special care has to be taken, then, that no samples are too similar not only between training and benchmark sets, but also between training and validation sets.

In the case of the validation set, here the samples are a random 5% of the total training set. This means

that samples in the validation set might be near-identical to those in the training set, making it hard to judge if the NN is overfitting the data at training time, since it will also implicitly overfit similar samples in the validation set.

In the case of the benchmark set, even though no two allele/peptide couples are identical with the training set, all alleles (except one) in the benchmark set are also used in the training/validation sets. Upon closer inspection of the data that was made available by the authors on Mendeley, I have also seen that for a 1000 random peptides from the “positive” samples in the benchmark set (“test.csv”), 353 of those were identical to “positive” peptides in the training set “train_hit.csv” and 429 were near-identical (i.e. the benchmark peptides were a sub-sequence of the training peptides). When doing the same test on “negative” peptides from the benchmark set, only six of those were identical to any “positive” peptide in the training set. It is not hard to imagine, then, that such an oversized NN might just recall identical or near-identical peptides (or near-identical allele/peptide couples) from the training set and still perform well on the “unseen” benchmark.

This also makes me doubt that MHCfovea’s good performance on rare alleles is trustworthy enough that we can use the NN to expand its usage to the larger set of 13k alleles, since an overfitting NN will usually give unexpected results on unseen data.

In order to ensure that MHCfovea’s performance is not result of overfitting, I suggest the authors to re-train and test it on new datasets so that:

- i. No peptides are identical or near-identical (e.g. one is a subsequence of the other) between training/validation set and benchmark set
- ii. No peptides are identical or near-identical between training set and validation set
- iii. No allele in the train set is identical to those in the validation set
- iv. No allele in the train set is identical to those in the benchmark

While I realize that points iii and iv might excessively limit the size of the train and/or benchmark sets, it would be possible to perform testing so that different versions of the NN are trained on different, non-overlapping subsets of train/benchmark data (see cross-fold or leave-one-out validation).

Minor issues:

- 2) Could the authors explain how they settled on this particular NN architecture? Have other architectures been explored, such as RNNs (which usually perform better when dealing with AA sequences)?
- 3) Why is the final activation of the NN a sigmoid and not a softmax, since this looks like a classification problem (prediction of binding/non-binding instead of, say, binding affinity)?

- 4) Related to the previous point: it seems to me like it would be incorrect to say that MHCfovea predicts binding probability, since the last activation is not a softmax, and the final prediction is the average of 18 different predictors. Please address or fix this.
- 5) Judging by the ScoreCAM paper, it seems like it should not be possible to apply it as is on MHCfovea (ScoreCAM is made for classification networks, and MHCfovea is not). Could the authors better specify how ScoreCAM was adapted to this particular architecture?
- 6) Please describe more in detail how the training set is split in downsized datasets (line 100 and following) since this was not immediately clear to me
- 7) Please state explicitly that the best ensemble was selected based on validation data (and not, say, on test data). This is implied in the text but I think it would be best to say it clearly
- 8) Line 185: “the allele signature was constructed from a subset of alleles in the cluster”, perhaps I missed this from somewhere else in the text, but how is this subset constructed?
- 9) Line 118: “less” should be “fewer”
- 10) Line 174: “meaning” should be “meaningful”

Reviewer #2 (Remarks to the Author: Reproducibility):

The evaluation code and full datasets have been made available by the authors and I was able to reproduce some of the results on this work. The training code is not available, so this work is not fully reproducible (i.e. generating similar NN models from the training data as well as the CAM scores)

Author rebuttal, first version:

Summary of revision

1. Benchmark data was partitioned into four groups: (1) unobserved alleles paired with dissimilar peptides; (2) unobserved alleles paired with similar peptides; (3) observed alleles paired with dissimilar peptides; and (4) observed alleles paired with similar peptides. Predictors were compared by different groups. (Results - Performance evaluation of MHCfovea’s predictor, Fig. 2f, Supplementary Figure 3h, Supplementary Table 5, and Supplementary Data 5)
2. Comparison of the training data of different predictors was added. (Results - Performance evaluation of MHCfovea’s predictor, Supplementary Figure 2, and Supplementary Table 2)

3. The list of unobserved alleles shared among different predictors was provided, along with the performance comparison based on these 10 alleles. (Results - Performance evaluation of MHCfovea's predictor, Fig. 2e, Supplementary Fig. 3e-g, Supplementary Table 4, and Supplementary Data 4)
4. An artificial data was created to demonstrate that MHCfovea does not tend to predict observed peptides as positives. (Discussion and Supplementary Figure 9)
5. ScoreCAM was applied on peptide sequences to show its capability of highlighting important positions. (Results - Selection of important MHC-I residues, Fig. 3a, and Supplementary Data 6)
6. The presentation of hyper-motifs and motifs was modified to provide better visualization quality. (Methods - Hyper-motifs and allele signatures, Fig. 4-5, Supplementary Figure 6-7, and Supplementary Data 8)
7. Descriptions about the hyperparameters of MHCfovea were added. (Methods - Model training and Supplementary Table 7)
8. %rank of the predicted scores was added as the output. (Methods - Threshold and %rank for the prediction score)

Response of "Connecting MHC-I-binding motifs with HLA alleles via deep learning"

Manuscript assessment and recommendation	
Editor's summary of the manuscript and overall assessment	Here, the authors have developed MHCfovea, a deep learning tool that can predict MHC-I binding motifs. When comparing AUC values, MHCfovea significantly outperforms three related tools: NetMHCpan4.1 (which also relies on deep learning), MHCflurry2.0, and MixmHCpred2.1. One benefit of this study is that the authors have also used MHCfovea to predict potential binding motifs for ~13,000 alleles, and summarize their results in an accessible online database. Both reviewers acknowledge that MHCfovea may present several advantages over existing tools, but raise concerns over the suitability of the training and input data sets used for benchmarking. There were also concerns about whether ScoreCAM is appropriately integrated into MHCfovea. While we would ultimately require the complete code to be made publicly available at the time of publication, we believe it is critical for the training code to be provided to reviewers in a revised manuscript. In summary, a resubmission should include, at a minimum, the following revisions: (1) Benchmark the performance of MHCfovea to related tools using the same training data set from other studies (as suggested by Referee #1) and update training and input data sets (as suggested by Referee #2).(2) Elaborate on the limitations of MHCfovea compared to other tools, as well as the lack of biological validation.(3) Justify the inclusion and suitability of ScoreCAM to MHCfovea, or

	remove it from the analytical pipeline, as mentioned by both reviewers. (4) Provide training code for reviewer evaluation, as mentioned by Referee #2. A late report was attached in the email of 4/20/2021. Please include responses to the 3rd reviewer in your rebuttal.
Response	In this revision, the manuscript was revised accordingly as follows: (1) Comparison of the training datasets between different predictors is provided in this revision in Supplementary Table 2 and Supplementary Fig. 2. The statistics reveal that MHCfovea has no obvious advantages over NetMHCpan4.1 and MHCflurry2.0 in terms of the number of alleles and peptides. (2) The lack of public training dataset and biological validation set on unobserved alleles for comparing predictors are noted as our limitation in the section of Discussion. (3) In this revision, we applied ScoreCAM on the peptide sequences, of which the anchor positions are well known, to demonstrate that ScoreCAM is capable of highlighting important positions in a classification problem (Fig. 3a). (4) All the source codes related to this study have been provided on GitHub (https://github.com/kohanlee1995/MHCfovea). (5) The responses to the comments of the 3rd reviewer are also included in this revision.

Reviewer #1

Overview		
Lee et al. present a novel method, MHCfovea, for predicting peptide-binding by MHC molecules and inferring peptide motifs and MHC allele signatures using deep learning. MHCfovea addressed the imbalanced classification of MHC-peptide binding prediction by taking advantage of the ensemble strategy with down-sampling. In the evaluation with independent data of eluted ligand, MHCfovea outperformed other methods. MHCfovea detected important positions in MHC sequences using ScoreCAM. They detected multiple binding motifs, and inferred meaningful hypermotifs with allelic signatures by clustering binding motifs. The manuscript is well written and convincing the value of their method. However, there are several points that should be addressed.		
Major concerns for revision		
#	Reviewer comment	Editorial comment
1	The improvement of prediction accuracy in prediction of peptide-binding is valuable. Could you describe more about whether the improvement is due to just the difference in a dataset used in training or any advantages in the algorithm? Which previous methods were trained on the current data set? If the improvement is probably due to just the difference in datasets, it should be noted as such.	It would be critical to confirm that MHCfovea outperforms related tools regardless of the training data set.
R	Thank you for the advice. This is also the concern of Reviewer #3 in comment 2. In this study, we selected The Immune Epitope Database (IEDB), the largest and well-known freely available resource, for experimental measurements related to immunopeptidome,	

as the data resource, which is also the data used by most of the other prediction algorithms, including NetMHCpan, MHCflurry, and MixMHCpred.

When compared with other methods in this study, the testing data (benchmark) is the same, but the training data is not. The training dataset of NetMHCpan4.1, the latest version of NetMHCpan, is not available, so we followed its method to build our training dataset in order to reduce the difference between datasets. The numbers of HLA alleles used as the experimental measurements for different methods are listed below (**Supplementary Table 2**). Similarly, the dataset of another famous algorithm, MHCflurry2.0, was also derived from data available on IEDB. Therefore, the procedures of constructing the training data of NetMHCpan4.1, MHCflurry2.0, and MHCfovea are considered consistent, although the numbers of alleles are different. When the number of alleles covered by the training data is considered, MHCfovea does not have advantages over NetMHCpan4.1 and MHCflurry2.0.

Supplementary Table 2. Summary of the training dataset used in each predictor

	MHCfovea	NetMHCpan	MHCflurry	MixMHCpred
# of HLA alleles	150	158	172	67
# of experimental measurements	375,802	698,566	522,132	252,165
# of decoys	20,609,534 (90x)	10,066,567	33,178,365	0
	6,869,853 (30x)			

The figure below (**Supplementary Fig. 2a**) is the Venn diagram of the alleles used in the training data of four different algorithms. MixMHCpred, which was developed in 2017, has the fewest alleles because of the lack of experimental measurements from mass

spectrometry at that time. In total, 148 of 150 (99%) alleles used in MHCfovea are overlapped with those in NetMHCpan and MHCflurry, and the two left alleles are overlapped with MHCflurry. Compared to MHCflurry, MHCfovea has no extra alleles but MHCflurry has 24 alleles which are unobserved in MHCfovea. When compared to NetMHCpan, MHCfovea has 2 alleles that were not in NetMHCpan, while NetMHCpan has 13 alleles that are unobserved in MHCfovea. In summary, in the allele level, MHCfovea has no obvious advantage over NetMHCpan and MHCflurry on the positive data of the training sets.

Supplementary Fig. 2

In the peptide level, both NetMHCpan4.1 and MHCflurry2.0-variant use much more experimental measurements than MHCfovea's. **Supplementary Fig. 2b** shows the overlap of experimental measurements used in MHCfovea, NetMHCpan4.1, and MHCflurry2.0. Only one peptide is unique in MHCfovea.

As for the negative peptides, we followed the method of NetMHCpan4.1 to prepare the decoy dataset. Different predictors have different D-E ratios in their training dataset. We trained our model on the D-E ratio of 30, 60, 90 in the training dataset to evaluate the effect of D-E ratio. It is probable that the large size of decoys contributes to the good

performance of MHCfovea. For comparison with NetMHCpan4.1, we listed the performance of the model with D-E ratio of 30 below (**Supplementary Table 3**). However, with fewer experimental measurements and decoys, MHCfovea still outperformed NetMHCpan4.1 and MHCflurry2.0.

Supplementary Table 3. Performance on the benchmark dataset

	AUC	AUC0.1	AP	PPV
MHCfovea (90x)	0.977	0.892	0.841	0.789
MHCfovea (30x)	0.977	0.892	0.832	0.780
NetMHCpan4.1	0.958	0.859	0.825	0.783
MHCflurry2.0	0.960	0.825	0.740	0.710
MixMHCpred2.1	0.942	0.823	0.767	0.723

In conclusion, we consider that MHCfovea has no obvious advantages over NetMHCpan4.1 and MHCflurry2.0 in terms of the number of alleles and peptides. On the other hand, the superior performance of MHCfovea over MixMHCpred2.1 might be owing to the advantage on the training data.

The related content was provided in the revised manuscript (**Result - page 6, line 115; Discussion - page 16, line 328**).

2

The idea of outputting probability scores is good; however, I wonder if there is a downside to not outputting %rank, which is commonly used in other tools. What is the practical threshold for the score that

If feasible, please include %rank as an output of MHCfovea. At a minimum, the use of probability scores

	shows significant bindings, and is it validated?	should be fully justified for consideration at Communications Biology.
R	It is very informative to add %rank in the prediction output which can explain the score more pellucidly than the probability only. We use 10,000 random peptides extracted from the UniProt database as the background to calculate the %rank of a prediction score by allele. When a prediction receives a %rank of 0.5, it means a peptide binds to MHC-I more probably than 99.5% random peptides. As for the threshold of positive prediction, setting a threshold at 0.68 can reach a maximal F1 score of 0.837 on the validation dataset. The related content can be found in the revised manuscript (Results - page7, line 128; Methods - page 22, line 443).	
3	It is interesting to evaluate the relationship between the degree of polymorphism and the sensitivity maps derived from ScoreCAM. However, I am not familiar with ScoreCAM, but I slightly wonder if the gradient is likely to be large where many categorical variables can be taken due to the original nature of the algorithm.	This comment is also relevant to point #11 from Referee #2. A revised manuscript should either further justify the inclusion of ScoreCAM, or remove it entirely from the MHCfovea pipeline.
R	ScoreCAM is a kind of class activation map (CAM). Unlike other CAM-based algorithms such as GradCAM, which is based on gradients, ScoreCAM depends on the weight of each activation map through its forward passing scores, which can highlight important regions and avoid noises more precisely. CAM can apply on a multi-class classification model, but different classes are based on different nodes of the final layer of the model. The prediction model of MHCfovea is a binary classification model whose final output layer has only one node representing the	

probability of positive prediction. Therefore, only CAM for the positive class was adopted in MHCfovea, and we selected extremely positive (prediction score > 0.9) peptides to highlight the important region of the positive class.

To reveal the effectiveness of ScoreCAM, in the revised manuscript, we applied it on the epitope part, which has been recognized that the second and last positions are anchor positions. The following figure (**Fig. 3a**) depicts the clustering heatmap of the peptide mask score per position and per allele. Only the first four and the last four positions are taken into consideration like the method used in constructing the peptide motifs. Mostly, the anchor positions, the second and last residues, have higher mask scores than other residues, demonstrating that ScoreCAM is capable of highlighting important positions in the peptide sequence.

Fig. 3a

Response Fig. 1

Here, we further examine the polymorphism of each position in the binding peptides, which are [2.762, 2.514, 2.836, 2.756, 2.818, 2.841, 2.77, 2.054] with respect to the positions [1, 2, 3, 4, -4, -3, -2, -1] (**Response Fig. 1**). We observed that the second and the

	last positions have considerably lower polymorphism, revealing that ScoreCAM does not tend to highlight the polymorphic positions as important positions. In summary, even though Fig. 3c reveals a positive correlation ($r=0.67$, $p=1.06e-11$) between importance from ScoreCAM and polymorphism, it is not always true. It is reasonable that higher polymorphic regions determine the pattern of epitopes more probably. With the demonstration of applying ScoreCAM on the peptide part, we consider that ScoreCAM is an appropriate method to highlight the important positions recognized by the CNN model. The related content was added in the revised manuscript (Results - page 9, line 174; Methods - page 23, line 474; Fig. 3a).	
4	The method of tuning hyper-parameters of the deep learning model should be written. Which data were used and what optimization method was used?	For the sake of reproducibility, it would be important to clearly state training data sets and optimization methods.
R	Thank you for the suggestion. There are several hyperparameters, including the training epochs, weight decay of Adam optimizer, batch size, and learning rate. We fixed the weight decay at 10^{-4} and the epochs at 30. The best model state was chosen after epoch 24 via the performance of the validation dataset to avoid overfitting, which was the reason we did not fine-tune the epochs. As for the batch size and learning rate, we used a grid search as the optimization method based on the average precision (AP) of the validation dataset. The batch size of 32 from the range [16, 32, 64] and the learning rate of 10^{-4} from the range [10^{-5}, 10^{-4}, 10^{-3}] were selected (Supplementary Table 7). The related descriptions were added in the Methods section (page 21, line 423; Supplementary Table 7).	

Minor concerns for revision		
#	Reviewer comment	Editorial comment
5	In page 8, the first sentence of the third paragraph of “Selection of important MHC-I residues”, I cannot understand why “non-polymorphic” positions were chosen as important positions. Isn’t it “polymorphic”?	
R	This was a typo. It should be “polymorphic”. The related content was added in the Results section (page 10, line 202). “Polymorphic positions with importance more than 0.4 were chosen as important positions. Fig. 3d presents the Venn diagram of position selection.”	
6	In page 9 and 10, the third sentence of the first paragraph and the fourth sentence of the third paragraph, the knowledge that the second and last positions of residues are important might be general; however, should be supported by citing a literature.	
R	Thanks for the advice. We have added the following references related to the anchor positions of epitopes (page 11, line 215) and the binding structure of MHC-I (page 12, line 255).  ● Bassani-Sternberg, M. & Gfeller, D. Unsupervised HLA Peptidome Deconvolution Improves Ligand Prediction Accuracy and Predicts Cooperative Effects in Peptide–HLA Interactions. J. Immunol. 197, 2492–2499 (2016). ● van Deutekom, H. W. M. & Keşmir, C. Zooming into the binding groove of HLA molecules: which positions and which substitutions change peptide binding most? Immunogenetics 67, 425–436 (2015). 	
7	Some of the figures may be difficult to understand because of missing labels and legends (e.g. horizontal axis of Fig. 3a, dot legend of Fig. 3b (a round dot is all residues?), and axis and heatmap label of Fig 5a).	Please update Figures 3/5 with additional definitions and graphical keys, as suggested.

		In general, we would recommend uploading individual image files with a revised manuscript to improve figure resolution. Please note that our formatting style ultimately requires the legend to clearly define all abbreviations, symbols, colors, and shading. Please write out the symbols/colors in words (blue circles, red dashed line, etc.) within these definitions.
R	We apologize for the unclear label of these figures. We have added the label and legend to Fig. 3 and Fig. 5.	
8	Some abbreviation should be decoded the first time they are mentioned in the main text (e.g. AUC, PR).	
R	We have decoded the abbreviation in the Results section (page 6, line 109). “The best model was that with D-E ratios of B=5 and A=90, respectively, showing an average precision (AP) of 0.898 and an area under the receiver operating characteristic curve (AUC) of 0.991.”	

Comments on impact, strength of the claims, and reproducibility		
#	Reviewer comment	Editorial comment
9	Impact of the study. The implementation of the algorithm is good, but the idea may not be fully novel. A study of predicting peptide-binding of MHC molecules using deep learning is now commonplace	The fact that this area has been explored before with machine learning methods reduces the impact and enthusiasm for publishing in Nature Communications. Altogether, the novelty and impact is not sufficient for consideration at Nature Methods. However, Communications Biology believes MHCfovea would represent an advance over other algorithms, if additional benchmarking is included.
10	Strength of the claims. The novelty in this work seems to be (1) the improvement in accuracy probably due to the ensemble learning; (2) the detection of important positions of MHC molecules visualization of the trained model, and binding-motif exploration.	
11	Reproducibility. To strengthen reproducibility, in performance comparisons with the previous methods, using the same input data would be warranted.	As mentioned in Point #1, it would be essential to confirm that MHCfovea outperforms related tools regardless of the training data set, for consideration in Communications Biology.

		The level of performance advance once this retraining is factored in (also highlighted by Reviewer #2) was concerning for Nature Communications.
--	--	---

Reviewer #2

Overview		
This work introduces MHCfovea, a deep Neural Network (NN) to predict the interaction between couples of MHC-I alleles and peptides. Along the prediction process, MHCfovea also shows which amino acids in the MHC-I target are deemed of interest in the interaction. This information can be used to build allele- and peptid-specific signatures through unsupervised learning (clustering). The text clearly states the objectives of this study and is overall well written. The code, available on github, is well organised and easy to use for evaluation (the training code is not available). The performance of the predictor seems to exceed the state of the art.		
Specific comments		
#	Reviewer comment	Editorial comment
1	Impact: Although this is not the first NN-based MHC-I binding predictor, it seems to me like it is the first to include elements from deep learning frameworks that can be used to explain the predictor's choices and learn new information about the underlying biological process (i.e. which amino acids play a role in the binding). This seems like a good idea, especially in a moment where explainability in AI is very much an issue. However, I have to point out some important flaws in the methods of this study that undermine the credibility of its results and have to be addressed before the study can be published.	
Major concerns for revision		

2	The main issue is the way the train, validation and benchmark sets are built. Although the authors state in the text that no allele/peptide couple in the benchmark set is identical to any in the train/validation sets, I am not convinced that this is enough to avoid information leakage between the sets and, subsequently, to an overestimation of the performance of the predictor. Each NN in the ensemble is made of approximately 500k weights, while the training set is made of approximately of 260k positive (binding) samples. A NN of such size could easily overfit the positive data so that these samples are “memorized” in it, which would make it unreliable when presented with previously unseen alleles or peptides. Special care has to be taken, then, that no samples are too similar not only between training and benchmark sets, but also between training and validation sets.	These issues related to training and biases within the datasets and consequent impact on model performance were an editorial concern for Nature Communications.
R	Thank you for the suggestion. Regarding the concern about the overfitting and memorization of the training process, we consider that the size of the training data is sufficient with respect to the network size. The model is made of approximately 500k weights for the possible sequence of MHC-I (182 a.a.) and peptide (8-15 a.a.). Considering the number of MHC-I alleles (about 13,000) and the potential number of peptides (about 20^{10}), we built a large model to learn the patterns of MHC-I-peptide binding. Since the number of available experimental measurements is about 260k and most of them are positive data, we built a large number of decoys (20 million) as the negative data, in order to provide sufficient data for parameter training We understand that the relation between the number of weights and samples cannot fully justify the concern about memorization explicitly. Thus, two analyses were added in this revision to demonstrate that MHCfovea’s better performance over other predictors is not from network memorization. First, an artificial data was created to demonstrate that MHCfovea does not tend to predict observed peptides as positives. The details are described in the response of comment 4. Second, benchmark data was partitioned into four groups: (1) unobserved alleles paired with dissimilar peptides; (2) unobserved alleles paired with similar peptides; (3) observed alleles paired with dissimilar peptides;	

	and (4) observed alleles paired with similar peptides. The performance of different predictors was compared on the four groups (see the response of comment 6). In addition, the groups (1) and (2) were combined to evaluate the performance of MHCfovea on unobserved alleles (see the response of comment 5). Along with the responses to the comments 4, 5, and 6, we believe MHCfovea achieves good generalization in predicting allele-peptide binding.															
3	In the case of the validation set, here the samples are a random 5% of the total training set. This means that samples in the validation set might be near-identical to those in the training set, making it hard to judge if the NN is overfitting the data at training time, since it will also implicitly overfit similar samples in the validation set.															
R	We realized the concern about overfitting due to similar peptides in the validation set. The validation set was used to fine tune the hyperparameters and D-E ratios. The similarity between the training and validation set might weaken the generalization of the model or even cause overfitting. We examined the peptide similarity between the training and validation set, and observed that similar and non-similar (denoted as dissimilar), peptides are about equally distributed in the validation set. The following table reveals that both groups have similar performance, indicating that the learning process has well taken care of both similar and dissimilar groups. According to the table shown below, we consider that MHCfovea has no obvious overfitting on the similar samples in the validation set. Response Table. The performance of MHCfovea on the similar and dissimilar peptides of the validation dataset     # positive # negative AUC AUC0.1 AP PPV     similar 7,623 211,919 0.993 0.915 0.899 0.828   			# positive	# negative	AUC	AUC0.1	AP	PPV	similar	7,623	211,919	0.993	0.915	0.899	0.828
	# positive	# negative	AUC	AUC0.1	AP	PPV										
similar	7,623	211,919	0.993	0.915	0.899	0.828										

	dissimilar	6,402	177,975	0.990	0.927	0.901	0.843
4	In the case of the benchmark set, even though no two allele/peptide couples are identical with the training set, all alleles (except one) in the benchmark set are also used in the training/validation sets. Upon closer inspection of the data that was made available by the authors on Mendeley, I have also seen that for a 1000 random peptides from the “positive” samples in the benchmark set (“test.csv”), 353 of those were identical to “positive” peptides in the training set “train_hit.csv” and 429 were near-identical (i.e. the benchmark peptides were a subsequence of the training peptides). When doing the same test on “negative” peptides from the benchmark set, only six of those were identical to any “positive” peptide in the training set. It is not hard to imagine, then, that such an oversized NN might just recall identical or near-identical peptides (or near-identical allele/peptide couples) from the training set and still perform well on the “unseen” benchmark.						
R	It is true that there are about 32.8% of the positive peptides in the benchmark set identical to the positive peptides in the training set, while only 0.4% of the negative peptides in the benchmark set are positive in the training set. To resolve the concern, we created an artificial set of pairing the positive peptides in the training set with all the other alleles in the benchmark set. For example, peptide “IEEKLNEA ” and allele “B*40:01” was a positive pair in the training dataset. Then, several pairs of this peptide and other alleles from the benchmark set were created to build the new artificial dataset. As shown in Supplementary Fig. 9, the score distribution of artificial pairs was close to that of negative samples, demonstrating that MHCfovea’s predictor does not memorize the observed sequences. Supplementary Fig. 9 compares the distribution of this artificial group with two groups originally present in the benchmark, positive and decoy pairs. The distribution of the artificial dataset is close to the negative data in the benchmark, which means that the						

artificial pairs are recognized as negative samples mostly by the MHCfovea’s predictor. That is, MHCfovea’s predictor does not memorize the observed sequences. We hope this evaluation can resolve the concern about overfitting and memorization on similar peptides.

Supplementary Figure 9

The related content was added in the Discussion section of the revised manuscript (page 15, line 304; Supplementary Fig. 9).

5 This also makes me doubt that MHCfovea’s good performance on rare alleles is trustworthy enough that we can use the NN to expand its usage to the larger set of 13k alleles, since an overfitting NN will usually give unexpected results on unseen data.

R

This is also the concern of Reviewer #3 in the comment 1. In the previous version, we used 25 rare alleles with fewer than 100 experimental measurements in the training dataset to evaluate the performance of MHCfovea. In fact, these 25 rare alleles contain 16 unobserved alleles. It's a pity that we didn't explain this point clearly.

In this revision, we replaced the evaluation of rare alleles with real unobserved alleles. In the benchmark dataset, there are 16 unobserved alleles not appearing in the training dataset. We compared the unobserved alleles with the observed alleles in the benchmark dataset (**Fig. 2e**). There is no significant difference between the AUC of observed and of unobserved alleles, which infers that MHCfovea also has good performance on the unobserved alleles. Moreover, 10 of 16 are commonly unobserved alleles for all other predictors, and MHCfovea is slightly better than the other methods (**Supplementary Fig. 3f**). Therefore, based on the result, we expand the prediction to 13,008 alleles of MHC-I to summarize the binding pattern of alleles.

Fig. 2e

Supplementary Fig. 3f

Next, to examine that the better performance of unobserved alleles is not all contributed by memorization. We partitioned the benchmark data into four groups mentioned in Question 2. The results show that in the groups of unobserved alleles, the AUC of dissimilar peptides is slightly lower than that of similar peptides (**Fig. 2f**). However, MHCfovea still outperforms others in both similar and dissimilar groups of

unobserved alleles. Therefore, we might infer that the good performance on unobserved alleles is not all caused by memorization.

Fig. 2f

The related content was added in the **Results** section of the revised manuscript (page 7, line 137 and page 8, line 151; Fig. 2e-f, and Supplementary Fig. 3e-h).

6

In order to ensure that MHCfovea’s performance is not result of overfitting, I suggest the authors to re-train and test it on new datasets so that:

- a. No peptides are identical or near-identical (e.g. one is a subsequence of the other) between training/validation set and benchmark set
- b. No peptides are identical or near-identical between training set and validation set
- c. No allele in the train set is identical to those in the validation set
- d. No allele in the train set is identical to those in the benchmark

MHCfovea analysis should be repeated following these criteria, for consideration at Communications Biology.

R

Thanks for your suggestion. We agree that a strict method is needed to prevent overfitting and memory in the training process. However, if we remove similar peptides from the training dataset, only 75% positive peptides would be left. We anticipate that this will cause a significant effect on the performance on the benchmark dataset. In addition, if doing this, the number of samples (**Supplementary Table 2**) will be much fewer than that of NetMHCpan4.1 and MHCflurry2.0, which makes an unfair comparison.

Supplementary Table 2. Summary of the training dataset used in each predictor

	MHCfovea	NetMHCpan	MHCflurry	MixMHCpred
# of HLA alleles	150	158	172	67
# of experimental measurements	375,802	698,566	522,132	252,165
# of decoys	20,609,534 (90x)	10,066,567	33,178,365	0
	6,869,853 (30x)			

Therefore, we applied these suggested criteria on the benchmark dataset. As mentioned in Question 2, benchmark data was partitioned into four groups: (1) unobserved alleles paired with dissimilar peptides; (2) unobserved alleles paired with similar peptides; (3) observed alleles paired with dissimilar peptides; and (4) observed alleles paired with similar peptides. The performance of similar and dissimilar peptides on the validation dataset is shown in Question 3. The performance on all the groups of MHCfovea (**Fig. 2f and Supplementary Fig. 3h**) is based on the same ratio of positive to negative samples as the original benchmark dataset. The similar groups have better performance than the

dissimilar groups, and it is reasonable that similar data can be learned well by a deep learning model.

The results show that the performance of the dissimilar group is still competitive with other predictors. For example, MHCfovea has better AUC than the other methods (**Fig. 2f**), and MHCfovea outperforms MHCflurry2.0 and MixMHCpred2.1, and is competitive with NetMHCpan4.1 as for the metric of AP (**Supplementary Fig. 3h**).

Supplementary Fig. 3h

The related content was added in the **Results** section of the revised manuscript (**page 8, line 151, Fig. 2f, and Supplementary Fig. 3h**).

- 7 While I realize that points 6c and 6d might excessively limit the size of the train and/or benchmark sets, it would be possible to perform testing so that different versions of the NN are trained on different, non-overlapping subsets of train/benchmark data (see cross-fold or leave-one-out validation).

R	We agree that cross validation is also a convincing and efficient method to evaluate the performance of a model. However, owing to the need of an isolated benchmark dataset for comparison with other predictors, we adopted the entire dataset from Sarkizova et al¹ as our benchmark. Given that the data number plays a major role in the training process of a deep learning model, we applied the suggestion mentioned in Question 6 on the benchmark dataset. We split the benchmark dataset into similar and dissimilar groups to evaluate the effect of similar data on the model performance. The result has been shown in Question 6. We hope the analysis and explanation can resolve the concern about overfitting and memorization. 1. Sarkizova, S. et al. A large peptidome dataset improves HLA class I epitope prediction across most of the human population. Nat. Biotechnol. 38, 199–209 (2020).
---	--

Minor concerns for revision		
#	Reviewer comment	Editorial comment
8	Could the authors explain how they settled on this particular NN architecture? Have other architectures been explored, such as RNNs (which usually perform better when dealing with AA sequences)?	
R	Actually, we have tried an LSTM model, and the performance is slightly worse than the CNN model. There are some other reasons for choosing the CNN architecture. First, the sequence of either MHC-I alleles or peptides can be encoded in a constant length with padding. Second, MHC-I alleles have polymorphic patterns and peptides have anchor patterns, which are both suitable for pattern recognition by a CNN model. Third, the CNN architecture can be calculated parallelly, which is appropriate for such a large dataset with about millions of samples. Finally, there are several methods for model interpretation in a CNN architecture. In MHCfovea, we used ScoreCAM to highlight the important positions recognized by our CNN model.	

	Owing to these reasons, we used a CNN architecture as the deep learning model in MHCfovea.	
9	Why is the final activation of the NN a sigmoid and not a softmax, since this looks like a classification problem (prediction of binding/non-binding instead of, say, binding affinity)?	
R	Our model is a binary classification model for the prediction of binding or non-binding, and only has a single output in the final layer which represents the positive probability. Softmax function is like a generalization of sigmoid function, and is mostly used in the multi-class classification. Therefore, in MHCfovea, and we used sigmoid function as the activation function.	
10	Related to the previous point: it seems to me like it would be incorrect to say that MHCfovea predicts binding probability, since the last activation is not a softmax, and the final prediction is the average of 18 different predictors. Please address or fix this.	
R	We appreciate your comment. The CNN model is a binary classification and uses sigmoid function as the final activation function, so the output is a score within 0 and 1. Because the label of training data is composed of 0 for non-binding and 1 for binding, the output of the model represents the positive probability, which is also regarded as the binding probability of the MHC-I allele and peptide. In addition, our model is an ensemble model of 18 independent CNN models, so the output is the average of the output, binding probability, of these models. We took the average of the 18 binding scores to reflect the binding probability predicted by the ensemble model.	
11	Judging by the ScoreCAM paper, it seems like it should not be possible to apply it as is on MHCfovea (ScoreCAM is made for classification networks, and MHCfovea is not). Could the authors better specify how ScoreCAM was adapted to this particular architecture?	This comment echoes point #3 from Referee #2; the use of ScoreCAM should be justified in a revised manuscript.

R This is also the concern of Reviewer #1 in Point #3. ScoreCAM is made for classification networks. The predictor of MHCfovea is a binary classification model. Although the output of the predictor is a single score, the score is used to represent the probability of the positive class. Therefore, ScoreCAM can be applied on a binary classification model, but the limitation is that it can only be applied on the positive class instead of the negative class. On the contrary, if a model is a two-class classification with two outputs, one for positive class and the other for negative class, ScoreCAM can be applied on both positive and negative classes to highlight the important regions of both two classes.

To validate the correctness and appropriateness of ScoreCAM, we applied it on the epitope part which has been recognized that the second and last position are anchor positions. The following figure (Fig. 3a) depicts the clustering heatmap of the peptide mask score per position and per allele. Only the first four and the last four positions are taken into consideration like the method used in generating peptide motifs. Mostly, the anchor positions, the second and last residues, have higher mask scores than other residues, which infers that ScoreCAM is capable of highlighting important positions in the peptide sequence.

Fig. 3a

	In addition, Fig.3c reveals the positive correlation ($r=0.67$, $p=1.06e-11$) between importance from ScoreCAM and polymorphism. It is reasonable that higher polymorphic regions determine the pattern of epitopes more probably. Therefore, according to these evaluations, we consider that ScoreCAM is an appropriate method to highlight the important positions recognized by our CNN model. The related content was added in the revised manuscript (Results - page 9, line 174; Methods - page 23, line 474; Fig. 3a).	
12	Please describe more in detail how the training set is split in downsized datasets (line 100 and following) since this was not immediately clear to me	
R	We apologize for the ambiguity. There are two levels of the D-E ratio. One is the D-E ratio of the overall dataset and the other is the D-E ratio of each downsized dataset. The D-E ratio of the overall benchmark dataset is about 30, so we prepared three times as many as the D-E ratio for the overall training dataset to evaluate the effect of the number of decoys on the model performance. In total, we have training decoys with the D-E ratio of 90. As for the downsized dataset, the D-E ratio is used to evaluate the effect of the imbalance between two classes on the model performance. Each downsized dataset contains shared experimental data and some non-overlapping decoys. In other words, the ensemble model of MHCfovea contains 18 CNN models and each of them was trained on all the experimental data and decoys with D-E ratio of 5, so all the decoys ($18 \times 5 = 90$) were observed in the ensemble model. This is our revision in the Results section of the revised manuscript (page 6, line 106). “Of note, all experimental data were shared in each downsized dataset, and the decoys were non-overlapping between each downsized dataset to make sure all the decoys were used in the ensemble model eventually.”	

13	Please state explicitly that the best ensemble was selected based on validation data (and not, say, on test data). This point is implied in the text but I think it would be best to say it clearly	
R	Sorry for the unclear description of our method. Indeed, we selected the best ensemble model according to the performance of the validation dataset. This point has been clarified in the revised manuscript (page 6, line 111). “Therefore, we used the ensemble model with 18 (=90/5) CNN models (the best performance on the validation dataset) as the predictor of MHCfovea.”	
14	Line 185: “the allele signature was constructed from a subset of alleles in the cluster”, perhaps I missed this from somewhere else in the text, but how is this subset constructed?	
R	In the previous version, we described the method of sampling alleles in Methods (page 25, line 529). To clear up the confusion, in this revision we further clarified the explanation in the Results section (page 12, line 238): “Of note, in each cluster, 50 alleles from each HLA group were randomly sampled to construct the allele signature to reduce the imbalance between different HLA groups.”	
15	Line 118: “less” should be “fewer”	
R	Thanks for the suggestion. We have corrected this typo (page 7, line 137). “Next, the performance of our pan-allele model was carefully examined in the context of 16 unobserved alleles (with no experimental measurements in the training dataset), listed in Supplementary Table 4.”	
16	Line 174: “meaning” should be “meaningful”	
R	Thanks for the suggestion. We have corrected this typo (page 11, line 224). “When exploring the relation between HLA sequences and MHC-I-binding motifs/sub-	

	motifs, we noticed that the number of alleles in a cluster is too small to form meaningful signatures.”	
17	Reproducibility: The evaluation code and full datasets have been made available by the authors and I was able to reproduce some of the results on this work. The training code is not available, so this work is not fully reproducible (i.e. generating similar NN models from the training data as well as the CAM scores)	Please make this code immediately available for reviewer input. We would recommend adding this information to your current GitHub page.
R	We appreciate the encouraging comment. We have added the source codes of the training process and ScoreCAM process on GitHub (https://github.com/kohanlee1995/MHCfovea).	

Reviewer #3

Overview		
The paper describes a new deep-learning framework that aims to extract the binding motifs for the wide diversity of MHC alleles. While accurate predictors of MHC-I-peptide binding affinity already exist, the main novelty of this approach lies in its interpretability.		
Specific comments		
#	Reviewer comment	Editorial comment
1	The authors apply MHCfovea to unobserved MHC alleles and use the predictions in their summarization module. This assumes that the MHCfovea predictions on these alleles are valid, which has not been tested. Nevertheless, the summarization module is built on the assumption that the predictions have the same reliability as on the observed MHC alleles. This is a major caveat. The manuscript would profit immensely from experimental validation for some of these unobserved interactions. Barring that, generalization capabilities to unobserved MHC alleles could be gauged by separating the data into training and validation sets in a way that ensures that MHC alleles contained in the validation set are not present in the training data (strict split). The cross-validation performance on such a dataset could give an indication of how well MHCfovea generalizes to MHC alleles that were not observed during training.	While we agree that experimental validation would improve the impact of the study, this point would not be necessary for Communications Biology, and could instead be addressed as a limitation.
R	We appreciate the encouraging comment. This is also the concern of Reviewer #2 (comment 5). In the previous version, we used 25 rare alleles with fewer than 100 experimental measurements in the training dataset to evaluate the performance of MHCfovea. In fact, these 25 rare alleles contain 16 unobserved alleles. So, in this	

revision, we focused more on the performance of MHCfovea on the 16 unobserved alleles in the benchmark dataset.

In this revision, we replaced the evaluation of rare alleles with real unobserved alleles. In the benchmark dataset, there are 16 unobserved alleles not appearing in the training dataset. We compared the unobserved alleles with the observed alleles in the benchmark dataset (**Fig. 2e**). There is no significant difference between the AUC of observed and of unobserved alleles, which infers that MHCfovea also has good performance on the unobserved alleles. Moreover, 10 of 16 are commonly unobserved alleles for all other predictors, and MHCfovea is slightly better than the other methods (**Supplementary Fig. 3f**).

In conclusion, both analyses demonstrate that MHCfovea’s predictor can generalize to unobserved alleles very well.

Fig. 2e

Supplementary Fig. 3f

The related content was provided in the **Results** section of the revised manuscript (page 7, line 137).

2 The comparison to NetMHCpan4.1, MHCflurry2.0 and MixMHCpred2.1 is problematic, as these models were

This point echoes similar concerns from Referee #1,

	not trained on the same data as MHCfovea. To ensure an unbiased comparison, the authors should train MHCfovea on the training data used to train the competing models for the comparison, if that data is available.	and should be addressed in a revision for consideration at Communications Biology.															
R	This is also the concern of Reviewer #1 in comment 1. In this study, we selected The Immune Epitope Database (IEDB), the largest and well-known freely available resource, for experimental measurements related to immunopeptidome, as the data resource, which is also the data used by most of the other prediction algorithms, including NetMHCpan, MHCflurry, and MixMHCpred. When compared with other methods in this study, the testing data (benchmark) is the same, but the training data is not. The training dataset of NetMHCpan4.1, the latest version of NetMHCpan, is not available, so we followed its method to build our training dataset in order to reduce the difference between datasets. The numbers of HLA alleles used as the experimental measurements for different methods are listed below (Supplementary Table 2). Similarly, the dataset of another famous algorithm, MHCflurry2.0, was also derived from data available on IEDB. Therefore, the procedures of constructing the training data of NetMHCpan4.1, MHCflurry2.0, and MHCfovea are considered consistent, although the numbers of alleles are different. When the number of alleles covered by the training data is considered, MHCfovea does not have advantages over NetMHCpan4.1 and MHCflurry2.0. Supplementary Table 2. Summary of the training dataset used in each predictor     MHCfovea NetMHCpan MHCflurry MixMHCpred     # of HLA alleles 150 158 172 67   # of experimental measurements 375,802 698,566 522,132 252,165   			MHCfovea	NetMHCpan	MHCflurry	MixMHCpred	# of HLA alleles	150	158	172	67	# of experimental measurements	375,802	698,566	522,132	252,165
	MHCfovea	NetMHCpan	MHCflurry	MixMHCpred													
# of HLA alleles	150	158	172	67													
# of experimental measurements	375,802	698,566	522,132	252,165													

# of decoys	20,609,534 (90x)	10,066,567	33,178,365	0
	6,869,853 (30x)			

The figure below (**Supplementary Fig. 2a**) is the Venn diagram of the alleles used in the training data of four different algorithms. MixMHCpred, which was developed in 2017, has the fewest alleles because of the lack of experimental measurements from mass spectrometry at that time. In total, 148 of 150 (99%) alleles used in MHCfovea are overlapped with those in NetMHCpan and MHCflurry, and the two left alleles are overlapped with MHCflurry. Compared to MHCflurry, MHCfovea has no extra alleles but MHCflurry has 24 alleles which are unobserved in MHCfovea. When compared to NetMHCpan, MHCfovea has 2 alleles that were not in NetMHCpan, while NetMHCpan has 13 alleles that are unobserved in MHCfovea. In summary, in the allele level, MHCfovea has no obvious advantage over NetMHCpan and MHCflurry on the positive data of the training sets.

Supplementary Fig. 2

In the peptide level, both NetMHCpan4.1 and MHCflurry2.0-variant use much more experimental measurements than MHCfovea's. **Supplementary Fig. 2b** shows the overlap of experimental measurements used in MHCfovea, NetMHCpan4.1, and

MHCflurry2.0. Only one peptide is unique in MHCfovea.

As for the negative peptides, we followed the method of NetMHCpan4.1 to prepare the decoy dataset. Different predictors have different D-E ratios in their training dataset. We trained our model on the D-E ratio of 30, 60, 90 in the training dataset to evaluate the effect of D-E ratio. It is probable that the large size of decoys contributes to the good performance of MHCfovea. For comparison with NetMHCpan4.1, we listed the performance of the model with D-E ratio of 30 below (**Supplementary Table 3**). However, with fewer experimental measurements and decoys, MHCfovea still outperformed NetMHCpan4.1 and MHCflurry2.0.

Supplementary Table 3. Performance on the benchmark dataset

	AUC	AUC0.1	AP	PPV
MHCfovea (90x)	0.977	0.892	0.841	0.789
MHCfovea (30x)	0.977	0.892	0.832	0.780
NetMHCpan4.1	0.958	0.859	0.825	0.783
MHCflurry2.0	0.960	0.825	0.740	0.710
MixMHCpred2.1	0.942	0.823	0.767	0.723

In conclusion, we consider that MHCfovea has no obvious advantages over NetMHCpan4.1 and MHCflurry2.0 in terms of the number of alleles and peptides. On the other hand, the superior performance of MHCfovea over MixMHCpred2.1 might be owing to the advantage on the training data.

The related content was provided in the revised manuscript (**Result - page 6, line 115;**

	Discussion - page 16, line 328).	
3	Are MHCfovea sequence motifs restricted to the peptides used in the training? In other words, does the model generalize to peptides not contained in the training data? Moreover, do I understand correctly that MHCfovea clusters the peptides into only 32 different groups (pairs of hyper-motifs)? If yes, then I think this restriction should be discussed in the manuscript. Again, generalization capabilities to unobserved peptides could be gauged with a strict split cross-validation.	
R	Thanks for the suggestion. First, the motifs are not restricted to the peptides used in the training set. On the contrary, we used a part of the benchmark set (254,742 peptides) as unobserved peptides to generalize the prediction on all 13,008 alleles. Therefore, in this revision, benchmark data was partitioned into four groups: (1) unobserved alleles paired with dissimilar peptides; (2) unobserved alleles paired with similar peptides; (3) observed alleles paired with dissimilar peptides; and (4) observed alleles paired with similar peptides. Fig. 2f and Supplementary Fig. 3h show the performance on different groups of MHCfovea. For MHCfovea, the performance on the group without similar peptides (denoted as the dissimilar group) is competitive with other predictors, even though the similar groups have better performance than the dissimilar groups. This implies that MHCfovea has good generalization on not only unobserved alleles (comment 1) but also dissimilar peptides.	

Fig. 2f

Supplementary Fig. 3h

Second, the number of clusters is fixed once summarization is completed. In this study, some minor clusters with fewer than 50 alleles were neglected, and in the end 32 major

	clusters are presented in our summarization. Most alleles (12,919 in 13,008, 99%) belong to one N-terminal and one C-terminal cluster within these 32 clusters. If new alleles are appended in the future, the process of allele extension and summarization can be reperformed to generate a new set of clusters. We have added the related content into the revised manuscript (Result - page 8, line 151; Discussion - page 15, line 321).
4	Lines 163-166: “Since the length of epitopes ranges from 8 to 15 and the significant residues are usually located at the second and last positions, we focused on the first four (N-terminal) and last four (C-terminal) residues to construct an 8-amino acid-long motif for each allele. Supplementary Fig. 4 depicts the hierarchical clustering of the binding motifs of HLA-B alleles.” This paragraph makes it sound like 8-amino acid long motifs are extracted from the allele sequence, while based on Fig. 1, I guess that the sub-motifs are actually constructed from the N- and C terminus of the peptide sequence. Throughout the text (starting in the abstract), this differentiation between MHC allele sequence motifs and peptide sequence motifs should be made clearer.
R	Sorry for the unclear description of motifs. The sub-motif is constructed from the peptide sequence. The following is our revision in Results (page 11, line 212). “Since the length of epitopes ranges from 8 to 15 and the significant residues are usually located at the second and last positions, we focused on the first four (N-terminal) and last four (C-terminal) residues to construct an 8-amino acid-long motif for peptides bound by each allele.²⁶”

	Moreover, to clarify the differentiation between motifs, we replaced most “motif” with “binding motif” in the revision. “Binding motif” and “MHC-I-binding motif” are used to represent the motif of peptides because peptides are binding by MHC-I alleles. “Sub-motif” is used to represent the N-terminus or C-terminus of binding peptides, and “hyper-motif” represents a group of motifs or sub-motifs. As for MHC-I alleles, we used “allele signature” to represent the pattern of MHC-I sequence.	
5	Lines 144-152: “The activation maps derived from CAM-based approaches are not sharp enough; residues next to the real important residue could be highlighted simultaneously. This explains why some non-polymorphic positions also have high importance; therefore, before applying linear regression, we removed all non-polymorphic positions. Fig. 3b presents a Pearson’s correlation of 0.67 ($P < 0.05$) between polymorphism and importance, and reveals that highly polymorphic sites play a more important role in the predictor. Non-polymorphic positions with importance more than 0.4 were chosen as important positions.” This seems contradictory, excluding non-polymorphic positions for the linear regression with the argument that they result from residue highlighting of neighbouring positions, while in the next passage treating them as the important positions. Please clarify.	
R	Thank you for the correction. It is true that we selected important positions from polymorphic regions. The following is the revision in the Results (page 10, line 202). “Polymorphic positions with importance more than 0.4 were chosen as important positions.”	
6	Lines 201-206: “The noticeable residues of N- and C-terminal hyper-motifs are mostly located in the first half	

	and last half part of allele signatures respectively, which is consistent with the binding structure of MHC-I molecules. For example, the E-dominant cluster has noticeable residues in the first half part of the allele signature; these residues are highly conserved in not only different combinations but also the cluster, which enhances confidence of the key residues highlighted in the allele signature.” This result does indeed enhance confidence in the method, but it is hard to see in the current version of Fig. 5. I encourage the authors to improve readability of the sequence motifs in Fig. 5 in order to show this result more convincingly.	
R	Thanks for the suggestion. In this revision, we only preserved the positive part of the allele signature to improve the readability. Here is our revision in the manuscript - Methods (page 26, line 540): “The $ASM^{cluster}$ was defined as the positive part of the difference between $PPM^{cluster}$ and $PPM^{background}$ in equation (4)”	
7	The sequence motif representation in Figures 1, 4, 5 and 6 is not ideal, as it is hard to see any but the most frequent amino acids. I would suggest changing the normalization so that the full space in the boxes can be used or finding ways to make the boxes bigger. For Allele signatures, it is also unclear to me what the mirrored amino acid sequences denote.	
R	We apologize for the ambiguity of allele signatures. To improve the readability, we removed the negative part of the matrix. Consequently, the scales of the allele signature and the highlighted allele signature in Fig. 4-5 and Fig. 6 are equal in this revision.	

8	Why are important positions in the MHC sequence extracted with ScoreCAM, but for the peptide you simply use the first and last four positions? What positions does ScoreCAM suggest as important in the peptide?	
R	Thanks for the encouraging comment. It is also important to apply ScoreCAM on the binding peptides. In the revised manuscript, we added related content and discussions. Fig. 3a depicts the clustering heatmap of the peptide mask score per position and per allele. Only the first four and the last four positions are taken into consideration like the method used in generating peptide motifs. Mostly, the anchor positions, the second and last residues, have higher mask scores than other residues, which is consistent with the knowledge of the anchor positions located in the second and last residues. This also provides a strong support for applying ScoreCAM on MHC-I sequences. The related content was provided in the revised manuscript (Results - page 9, line 174; Methods - page 23, line 474; Fig. 3a).  Fig. 3a	
9	Lines 421-422: “In this study, we used PPM to calculate the MHC-I sequence motif and ICM to calculate the MHC-	

	I-binding motif.” Why two different methods?	
R	We realize the concern about the different methods on alleles and peptides. Due to different background frequencies for peptides and alleles, we used two different methods to represent the MHC-I sequence motif and MHC-I-binding motif. For binding motifs, which are constructed from a large set of decoy peptides, it is reasonable to use information content matrix (ICM) that takes the background frequency of amino acids into consideration. It is indeed that the frequency of each amino acid on each position is almost equal (≈ 0.05). In such a situation, the ICM serves as a good way to highlight the amino acid with frequency more than the background frequency. By contrast, for allele sequences, only real human allele sequences are present in the dataset, i.e. the background frequency for each amino acid on each position is not 0.05. The number of allele sequences is limited, and the frequency of each amino acid is not equal. Therefore, we used the difference between the motif of target alleles and the background motif to represent the sequence pattern of the target alleles. The position probability matrix (PPM) without an assumption of background frequency is suitable for the calculation of the difference between two matrices.	

Open Research Evaluation

Data Availability	
Data Availability Statement	This journal strongly supports public availability of data and custom code associated with the paper in a persistent repository where they can be freely and enduringly accessed or as a supplementary data file when no

	appropriate repository is available. If data and code can only be shared on request, please explain why in your data Availability Statement, and also in the correspondence with your editor. For more information, please refer to https://www.nature.com/nature-research/editorialpolicies/reporting-standards#availability-of-data Please ensure that datasets deposited in public repositories are now publicly accessible, and that accession codes or DOI are provided in the "Data Availability" section. As long as these datasets are not public, we cannot proceed with the acceptance of your paper. For data that have been obtained from publicly available sources, please provide a URL and the specific data product name in the data availability statement. Data with a DOI should be further cited in the methods reference section.
Mandatory data deposition	http://www.nature.com/authors/policies/availability.html#data
Source data	The following figure panels should be accompanied by the underlying source data: Fig. 2b-g, Fig. 3a-b, Fig. 6a-b, Supp. Fig. 2. Source data files (in Excel or text format) will be mandatory prior to publication in a Nature Portfolio journal.
Response:	Thanks for the advice. We have added Supplementary Data 3-5 for Fig. 2 and Supplementary Fig. 3, Supplementary Data 6-7 for Fig. 3, and Supplementary Data 10-11 for Fig. 6.
Data citation	Where datasets are hosted in public repositories that provide datasets with Digital Object Identifiers (DOIs), we encourage these datasets to be formally cited in reference lists. Citations of datasets, when they appear in the reference list, should include the minimum information

	recommended by DataCite and follow journal style. For example: Hao, Z., AghaKouchak, A., Nakhjiri, N., Farahmand, A. Global Integrated Drought Monitoring and Prediction System (GIDMaPS) Data sets. figshare. http://dx.doi.org/10.6084/m9.figshare.853801 (2014) Citing and referencing data in publications supports reproducible research, by increasing the transparency and provenance tracking of data generated or analysed during research. Citing data formally in reference lists also helps facilitate the tracking of data reuse and may help assign credit for individuals' contributions to research. A number of Springer Nature imprints are signatories of the Joint Declaration on Data Citation Principles, which stress the importance of data resources in scientific communication.
Data Publication Recommended	We suggest that you consider publishing your MHCfovea dataset as a Data Descriptor in Scientific Data. See the journal website for details. https://www.nature.com/sdata/
Code Availability	
Code Availability	As stated by Referee #2 (point #17), please provide training code for reviewer input. We would recommend adding this code to the existing Github page for MHCfovea.
Response:	Thanks for the suggestion. We have added the source codes of training process and ScoreCAM process on the GitHub (https://github.com/kohanlee1995/MHCfovea)

Code Citation	In addition to making the custom code available, we ask that you ensure that the version of the code/software described in the paper is deposited in a DOI-minting repository (eg, Zenodo) and that this DOI is also cited in the main Reference list.
Code Reporting	Please ensure that any software mentioned in the Reporting Summary (Python, numpy, etc.) are also listed in the manuscript.
Research ethics	
Research with human participants	Research involving human research participants must have been performed in accordance with the Declaration of Helsinki. Please provide confirmation that all relevant ethical regulations were followed and that informed consent was obtained. This must be stated in the Methods section, including the name of the board and institution that approved the study protocol. See our ethics policy for details. Please ensure that relevant ethics statements from source data sets are included in the revised manuscript.
Methods assessment and reproducibility	
Data Presentation	Please ensure that data presented in a plot, chart or other visual representation format shows data distribution clearly (e.g. dot plots, box-and-whisker plots). When using bar charts, please overlay the corresponding data points (as dot plots) whenever possible and always for $n \leq 10$. (Please see the following editorial for the rationale behind this request and an example https://www.nature.com/articles/s41551-017-0079). Large datasets must be supplied as separate Supplementary Data files. Each file must be labelled as Supplementary Data 1, etc. Please reclassify Supplementary Tables 2, 3, and 5 as Supplementary Data 1-3.

Response:	Thanks for the advice. We have reclassified these Supplementary Tables as Supplementary Data.
Statistical Evaluation	Wherever statistics have been derived (e.g. error bars, box plots, statistical significance) the legend needs to provide and define the n number (i.e. the sample size used to derive statistics) as a precise value (not a range), using the wording “n=X biologically independent samples/animals/cells/independent experiments/n=X cells examined over Y independent experiments” etc. as applicable. Please note that statistics such as error bars significance and p values cannot be derived from $n < 3$ and must be removed in all such cases. We strongly discourage deriving statistics from technical replicates, unless there is a clear scientific justification for why providing this information is important. Conflating technical and biological variability, e.g., by pooling technically replicates samples across independent experiments is strongly discouraged. (For examples of expected description of statistics in figure legends, please see the following https://www.nature.com/articles/s41467-019-11636-5 or https://www.nature.com/articles/s41467-019-11510-4).
Figure Legends	The figure legends must indicate the statistical test used. Where appropriate, please indicate in the figure legends whether the statistical tests were onesided or two-sided and whether adjustments were made for multiple comparisons. For null hypothesis testing, please indicate the test statistic (e.g. F, t, r) with confidence intervals, effect sizes, degrees of freedom and P values noted.

	Please provide the test results (e.g. P values) as exact values whenever possible and with confidence intervals noted. Please update the legends of Figures 2e-2g, 6a-6b, and Supplementary Figures 2a-2f, 2h, accordingly. All error bars need to be defined in the legends (e.g. SD, SEM) together with a measure of centre (e.g. mean, median). For example, the legends should state something along the lines of “Data are presented as mean values +/- SEM” as appropriate. All box plots need to be defined in the legends in terms of minima, maxima, centre, bounds of box and whiskers and percentile. Please update Fig. 6, accordingly.
Response:	Thanks for the advice. We have added the statistical analysis of these figures to the corresponding Supplementary Data.
Reproducibility	Please state in the legends how many times each experiment was repeated independently with similar results. This is needed for all experiments, but is particularly important wherever results from representative experiments (such as micrographs) are shown. If space in the legends is limiting, this information can be included in a section titled “Statistics and Reproducibility” in the methods section.
Other notes	
Attached Reporting Summary	We have included as an attachment to the decision letter a version of your Reporting Summary with a few notes. This is mainly for your information, but we hope it is helpful when preparing your revised manuscript. If you decide to resubmit the manuscript for further consideration at Communications Biology, please be sure to include an updated Reporting Summary.

Reviewer comments, second version:

REVIEWERS' COMMENTS:

Reviewer #1 (Remarks to the Author: Overall significance):

In this revision, the authors responded well to all the major concerns with appropriate figures and additional analyses. The comment for each of them is as follows.

- The concerns in the improvement of prediction accuracy have been well addressed. They showed that the improvement of MHCfovea over NetMHCpan4.1 and MHCflurry2.0 was not just due to the increase in the training data by comparing the numbers of training data in terms of alleles and peptides. The superiority over MixMHCpred2.1 might be partly due to the dataset; however, the improvement was obvious.
- They additionally implemented to output %rank. They also provided a threshold for the significant binding.
- They added an analysis focused on the epitope part to demonstrate the efficacy of ScoreCAM. They revealed that the second and the last positions, which have lower polymorphism, presented higher mask score. Although a rigorous mathematical proof might be difficult, it would be sufficient for the main purpose of this study to show the validity of ScoreCAM using a region of which importance is expected in advance.
- They appropriately provided the method of tuning hyper-parameters. They performed tuning fairly by using only training and validation data, without referring to the benchmark data.

All the minor points have been also well addressed.

Regarding Reviewer #3's comments

The authors addressed most of the points made by Reviewer#3 with convincing additional analyses and figures; but might not fully respond to the comment #8. However, the answer seems obvious to some extent and might not need revision in this point. My point-by-point comments are as follows.

#1. The authors addressed the concern on generalization capability for the unobserved MHC alleles by evaluating the accuracies for real unobserved alleles. The results are convincing in that MHCfovea achieved equivalent accuracies for unobserved alleles as observed alleles and performed better than other methods.

#2. Ideally, MHCfovea should have been trained with training data used in other methods for fair comparisons; however, they could not obtain the same datasets. Instead, they showed that the improvement of MHCfovea over NetMHCpan4.1 and MHCflurry2.0 was not just due to the increase in the training data by comparing the numbers of training data in terms of alleles and peptides.

#3. The authors addressed the concern on generalization capability for the peptide not included in the training data by separating the benchmark set accordingly. Although I could hardly understand why the accuracies for unobserved alleles were better than those for observed alleles, they showed that the performance for dissimilar peptides was competitive with that for similar peptides.

#8. I interpret that the Reviewer #3 asked why the authors limited the discussion of importance only to the first and last four positions despite that they could evaluate the full lengths of peptides. In other words, it seems that the Reviewer #3 was asking how important were the peptide positions other than first and last four positions. Although the authors showed that the second and last residues were most important among the first and last four positions, they might not fully respond to the question in this sense. However, I can understand that the peptides are basically from 8 to 15 length and it is reasonable to limit the evaluation to the first and last four positions for normalized motif discovery.

#4-7, 9. All the minor points have been well addressed.

Reviewer #2 (Remarks to the Author: Strength of the claims):

The authors have addressed each of the concerns I have raised in my previous review, especially when it comes to potential biases in the training and benchmark data, i.e. the fact that although a small bias was indeed present, there still is an improvement over previous methods and the good prediction performance holds on unobserved alleles (though this could not be validated experimentally).

Reviewer #2 (Remarks to the Author: Reproducibility):

I thank the reviewers for making the complete source code available on their git repository.

Overall, I believe this is a much improved manuscript that is suited for publication.

Author rebuttal, second version:

REVIEWERS' COMMENTS:

Reviewer #1 (Remarks to the Author: Overall significance)

In this revision, the authors responded well to all the major concerns with appropriate figures and additional analyses. The comment for each of them is as follows.

1. The concerns in the improvement of prediction accuracy have been well addressed. They showed that the improvement of MHCfovea over NetMHCpan4.1 and MHCflurry2.0 was not just due to the increase in the training data by comparing the numbers of training data in terms of alleles and peptides. The superiority over MixMHCpred2.1 might be partly due to the dataset; however, the improvement was obvious.
Response: We appreciate the comment which helped us clarify the confidence of accuracy improvement.
2. They additionally implemented to output %rank. They also provided a threshold for the significant binding.
Response: Thank you for the suggestion.
3. They added an analysis focused on the epitope part to demonstrate the efficacy of ScoreCAM. They revealed that the second and the last positions, which have lower polymorphism, presented higher mask score. Although a rigorous mathematical proof might be difficult, it would be sufficient for the main purpose of this study to show the validity of ScoreCAM using a region of which importance is expected in advance.
Response: We are glad that the added analysis reveals the efficacy of ScoreCAM. Thank you!
4. They appropriately provided the method of tuning hyper-parameters. They performed tuning fairly by using only training and validation data, without referring to the benchmark data.

All the minor points have been also well addressed.

Response: Thanks for all the comments and suggestions.

Regarding Reviewer #3's comments (commented by Reviewer #1)

The authors addressed most of the points made by Reviewer#3 with convincing additional analyses and figures; but might not fully respond to the comment #8. However, the answer seems obvious to some extent and might not need revision in this point. My point-by-point comments are as follows.

1. The authors addressed the concern on generalization capability for the unobserved MHC alleles by evaluating the accuracies for real unobserved alleles. The results are convincing in that MHCfovea achieved equivalent accuracies for unobserved alleles as observed alleles and performed better than other methods.
Response: Thanks for the advice. It is very important to validate our method on unobserved alleles.
2. Ideally, MHCfovea should have been trained with training data used in other methods for fair comparisons; however, they could not obtain the same datasets. Instead, they showed that the improvement of MHCfovea over NetMHCpan4.1 and MHCflurry2.0 was not just due to the increase in the training data by comparing the numbers of training data in terms of alleles and peptides.
Response: Thank you for the comment which helped us clarify the contribution of MHCfovea on accuracy improvement.
3. The authors addressed the concern on generalization capability for the peptide not included in the training data by separating the benchmark set accordingly. Although I could hardly understand why the accuracies for unobserved alleles were better than those for observed alleles, they showed that the performance for dissimilar peptides was competitive with that for similar peptides.
Response: The number of unobserved alleles in our benchmark is a small portion of MHC-I alleles. More data of unobserved alleles are needed to well validate the performance on unobserved alleles. However, this limitation has been noted in the discussion.
8. I interpret that the Reviewer #3 asked why the authors limited the discussion of importance only to the first and last four positions despite that they could evaluate the full lengths of peptides. In other words, it seems that the Reviewer #3 was asking how important were the peptide positions other than first and last four positions. Although the authors showed that the second and last residues were most important among the first and last four positions, they might not fully respond to the question in this sense. However, I can understand that the peptides are basically from 8 to 15 length and it is reasonable to limit the evaluation to the first and last four positions for normalized motif discovery.
Response: Thanks for the interpretation. Indeed, to provide a general summarization for possible peptides with length of 8-15 a.a., we limited the evaluation on the first and last four positions.

#4-7, 9. All the minor points have been well addressed.

Response: Thanks for all the comments.

Reviewer #2

Remarks to the Author: Strength of the claims

The authors have addressed each of the concerns I have raised in my previous review, especially when it comes to potential biases in the training and benchmark data, i.e. the fact that although a small bias was indeed present, there still is an improvement over previous methods and the good prediction performance holds on unobserved alleles (though this could not be validated experimentally).

Remarks to the Author: Reproducibility

I thank the reviewers for making the complete source code available on their git repository.

Overall, I believe this is a much improved manuscript that is suited for publication.

Response: We are glad the added analysis resolved the concerns regarding the similar peptides present in the training data. Thank you!